# Deficiency of a triterpene pathway results in humidity-sensitive genic male sterility in rice

Zheyong Xue[1], Xia Xu[1,2], Yuan Zhou[1,2], Xiaoning Wang[3], Yingchun Zhang[1], Dan Liu[1,2], Binbin Zhao[1], Lixin Duan[1,4] & Xiaoquan Qi [1]

In flowering plants, the pollen coat protects the released male germ cells from desiccation and damage during pollination. However, we know little about the mechanism by which the chemical composition of the pollen coat prevents dehydration of pollen grains. Here we report that deficiency of a grass conserved triterpene synthase, OsOSC12/OsPTS1, in rice leads to failure of pollen coat formation. The mutant plants are male sterile at low relative humidity (RH < 60%), but fully male fertile at high relative humidity (>80%). The lack of three major fatty acids in the pollen coat results in rapid dehydration of pollen grains. We show that applying mixtures of linolenic acid and palmitic acid or stearic acid are able to prevent over-dehydration of mutant pollen grains. We propose that humidity-sensitive genic male sterility (HGMS) could be a desirable trait for hybrid breeding in rice, wheat, maize, and other crops.

[1] Key Laboratory of Plant Molecular Physiology, Institute of Botany, Chinese Academy of Sciences, Nanxincun 20, Fragrant Hill, Beijing 100093, China. [2] University of Chinese Academy of Sciences, Yuquan Road 19, Beijing 100049, China. [3] Department of Natural Product Chemistry, Key Laboratory of Chemical Biology (Ministry of Education), School of Pharmaceutical Sciences, Shandong University, 44 West Wenhua Road, Jinan 250012, China. [4] Present address: International Institute for Translational Chinese Medicine, Guangzhou University of Chinese Medicine, Guangzhou, 510006 Guangdong, China. Zheyong Xue, Xia Xu and Yuan Zhou contributed equally to this work. Correspondence and requests for materials should be addressed to X.Q. (email: xqi@ibcas.ac.cn)

Conditional male sterility systems facilitate breeding of hybrid varieties with yield advantages in many crops[1]. Photoperiod-/thermo-sensitive genic male sterility (P/TGMS)[2,3] that is regulated through a non-coding RNA or small RNA[4–6] has been used to generate hybrid varieties of rice (*Oryza sativa* L.) using a two-line hybrid system, which account >20% of the total planting area of hybrid rice in China[7]. However, the sterility may sometimes be compromised by unpredictable fluctuations in temperature[8]. To more fully exploit the potential of two-line hybrid breeding systems, additional conditional male sterility systems could be beneficial.

The pollen coat, the outermost layer of the pollen grain, protects the released pollen grain from desiccation, damage, and pathogen attack, enabling the completion of pollination[9,10]. Chemical analysis of the pollen coat revealed the presence of carotenoids, flavonoids, fatty acids, isoprenoids, glycoconjugates, and many other compounds[11–13]. Rice and other grass species appear to have less thinner pollen coats materials than have been described in Brassicaceae species[14–17]. Mutants that are deficient in pollen coat formation may have conditional male sterile phenotypes[18].

The constituents of pollen coats include isoprenoids. Oxidosqualene cyclases (OSCs) represent a branch point enzymes in isoprenoid biosynthesis, converting a basic terpenoid precursor, 2,3-oxidosqualene, to a diverse set of functional steroids and triterpenoids, which are often involved in plant defence[19–22]. OsOSC12 is a grass-species-specific triterpene synthase that is expressed in the anther[22].

In this study, we show that OsOSC12 is a bicyclic triterpene synthase and catalyzes a commited step in a triterpene pathway.

OsOSC12-deffective mutants have less pollen coat material than wild type (WT) and display a humidity-sensitive genic male sterility (HGMS) phenotype.

## Results

**Tapetum-specific expression of OsOSC12.** Among the annotated 12 rice OSCs[21], OsOSC12 is specifically expressed in the anther[22]. OsOSC12 and its encoded protein are expressed in tapetal cells at relatively late stages of anther development[14,15] from S8 to S11 (Fig. 1 and Supplementary Figs. 1–3). During S10 and 11, as the tapetal cells start to degrade, the lipid bodies are released into the anther lumen and relocate to form an extracellular lipidic coating around maturing pollen grains[11,14,15]. We therefore hypothesized that OsOSC12 is involved in pollen development, specifically in pollen wall formation.

**OsOSC12-defective mutants are HGMS.** We identified six mutants which carry nonsense or missense mutations for OsOSC12 by screening ethyl methane sulfonate (EMS) or sodium azide-induced rice mutant populations by Targeting Induced Local Lesions IN Genomes (TILLING) method (Fig. 2a and Supplementary Table 1). The E157 mutation induced a premature stop codon and was associated with both a substantial reduction of OsOSC12 transcript abundance and OsOSC12 protein in the anther (Fig. 2b, c and Supplementary Fig. 4). Under normal field conditions (30–39 °C, relative humidity; RH 40–70%), the level of self fertility shown in E157, S1708, and S4928 plants was reduced to <6% of WT plants, while E2304, S558, and S1535 plants were all fully self-fertile (Fig. 2d). Genetic analysis (Supplementary

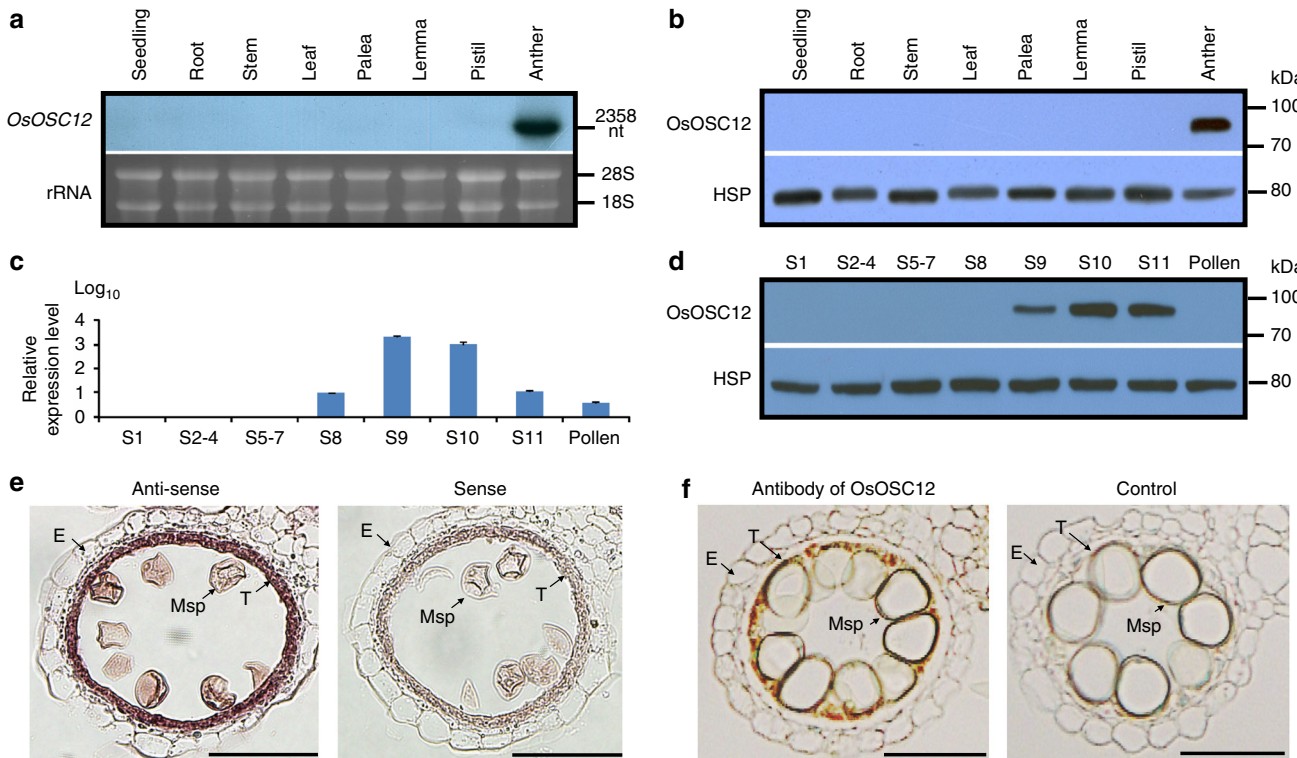

**Fig. 1** Transcription and expression of *OsOSC12*. **a**, **b** Tissue specificity as revealed by northern and western blotting. **c**, **d** Temporal profiles during anther development, as shown by real-time PCR and western blotting. S1 to S11 correspond to the stages of anther development as described[15]. The values are presented as means ± s.e., n = 3. **e** The localization of *OsOSC12* mRNA at S9. Hybridization with sense *OsOSC12* transcript provided the negative control. **f** Immuno-localization of OsOSC12 at S10. Hybridization with blocking buffer was used as the negative control. The negative controls for northern and western blotting experiments were, respectively, 18S and 28S rRNA, and the rice heat shot protein Q69QQ6. E epidermis, Msp microspore, T tapetum. Scale bar, 50 µm

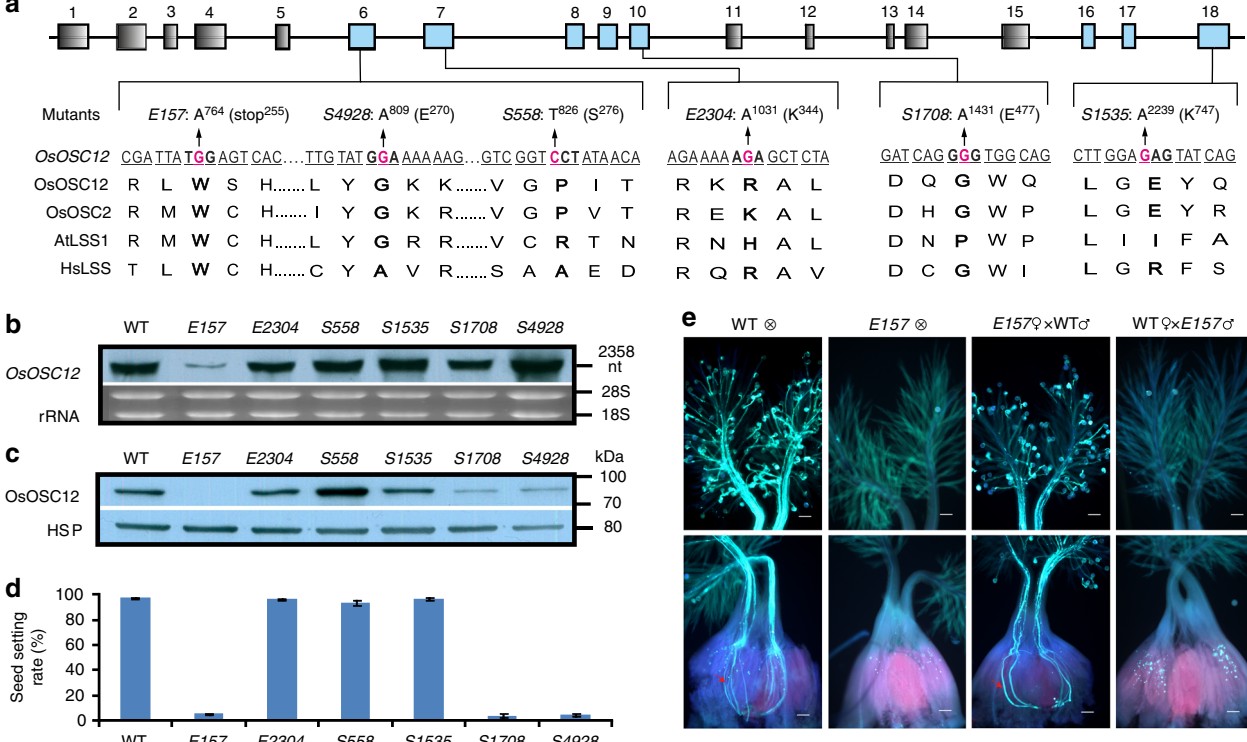

**Fig. 2** The absence of OsOSC12 results in male sterility. **a** *OsOSC12* mutants induced by either EMS or sodium azide. Boxes and lines indicate the *OsOSC12* exons and introns, respectively. TILLING was directed at exons 6–10 and 16–18 (blue box). Mutated nucleotides are marked in red, and the changed nucleotides and the corresponding amino acids are shown for each mutant. **b** Northern blot analysis of *OsOSC12* transcription. **c** Western blot analysis of OsOSC12 expression. Negative controls as given in Fig. 1. **d** Self fertility of field-grown WT and mutant plants (30–39 °C, 40–70% RH), based on panicles. The data are presented as mean ± s.d. *n* = 15. **e** Germination (upper panel) and pollen tube elongation (lower panel) of WT and *E157* pollen, imaged 1 h after pollination. Scale bar, 100 μm

Table 2) indicated that sterility of mutants *E157*, *S1708*, and *S4928* was due to recessive mutations of *OsOSC12*. The allelism test (Supplementary Fig. 5) showed that the *E157*, *S1708*, and *S4928* mutations all represented loss-of-function *OsOSC12* alleles. WT pollen germinated freely on both WT and *E157* stigmas, but *E157* pollen grains rarely adhered to the stigma and germinated poorly at normal RH (30–60%) (Fig. 2e). All loss-of-function mutants *E157, S1708*, and *S4928* plants became male fertile when complemented by transformation with the WT *OsOSC12* allele (Supplementary Fig. 6). Thus we conclude that mutations of *OsOSC12* cause male sterility.

When pollen grains were placed on a detached mature pistil under an RH of 30–60%, WT pollen adhered to the stigma after 20.6 ± 6.2 s (*n* = 178) and thereafter developed normally (Fig. 3a), but *E157* pollen began to shrink after 18.5 ± 10.6 s (*n* = 103), becoming dehydrated by 48.3 ± 10.4 s (*n* = 103) (Fig. 3a). As a result, the pollen did not adhere to the stigma and pollen tube elongation was arrested at a very early stage (Supplementary Fig. 7). Exposing the dehydrated *E157* pollen to an RH above 80% restored both their original shape and their ability to adhere (Fig. 3a and Supplementary Movie 1) and germinate (Supplementary Fig. 7). There was no difference between the ability of *E157* and WT pollen to germinate under the conditions of high RH. Thus, OsOSC12 pertains to pollen dehydration, but is not required for pollen recognition, adhesion, or germination.

When freshly shed pollen grains were placed on a glass slide and maintained at 27–32 °C and 30–60% RH, *E157*, *S1708*, and *S4928* grains shrank immediately, becoming fully dehydrated within a minute, while the WT pollen grains adhered to one another, produced exudate and shrank over a period of about 30

min (Fig. 3b). Seed setting rates of the three mutants were <2% when exposed to an RH of 30–60%, but was >80% when the RH was raised to above 80% (Fig. 3c, d). This conditional HGMS is due to dehydration and rapid loss of pollen viability at ambient or low relative humidity (RH < 60%). This is different from the *Arabidopsis pop1* mutant, which loses the ability to absorb water from the stigma[18].

**Three fatty acids prevent pollen dehydration.** Transmission electron microscopy (TEM) observation of ultramicrotome and cryo-ultramicrotome pollen sections showed that in contrast to WT, *E157* pollen lacked pollen coat material deposited between the tectum and the nexine after anther stage S12 (Fig. 4a–c and Supplementary Figs. 8, 9). It is possible that the lack of this pollen coat materials may cause rapid dehydration of the mutant pollen. To further decipher the chemical constitution of the pollen coat, we developed a method to extract pollen coat compounds (Supplementary Fig. 10), and obtained about 4% (w/w) pollen coat extract (PCE, 0.2 g) from 5 g pollen grains. Gas chromatography–mass spectrometry (GC–MS) analysis of PCE revealed that long-chain fatty acids, sterols, and several unknown peaks in the region of triterpene esters, were substantially reduced in *E157* compared to the WT, while three major phytosterols increased in the mutant *E157* (Supplementary Fig. 10). Palmitic acid (16:0) and linolenic acid (18:3), which comprise 3.4 ± 0.23% (mean ± s.d., *n* = 5) and 6.9 ± 0.11% (mean ± s.d., *n* = 5) of the WT PCE (w/w), respectively, were decreased 4.93 and 4.51-fold (*P* < 0.01 by Student's *t* test) in *E157* PCE (Fig. 4d). Stearic acid (18:0), which comprised about 2.8 ± 0.11% (mean ± s.d., *n* = 5) of

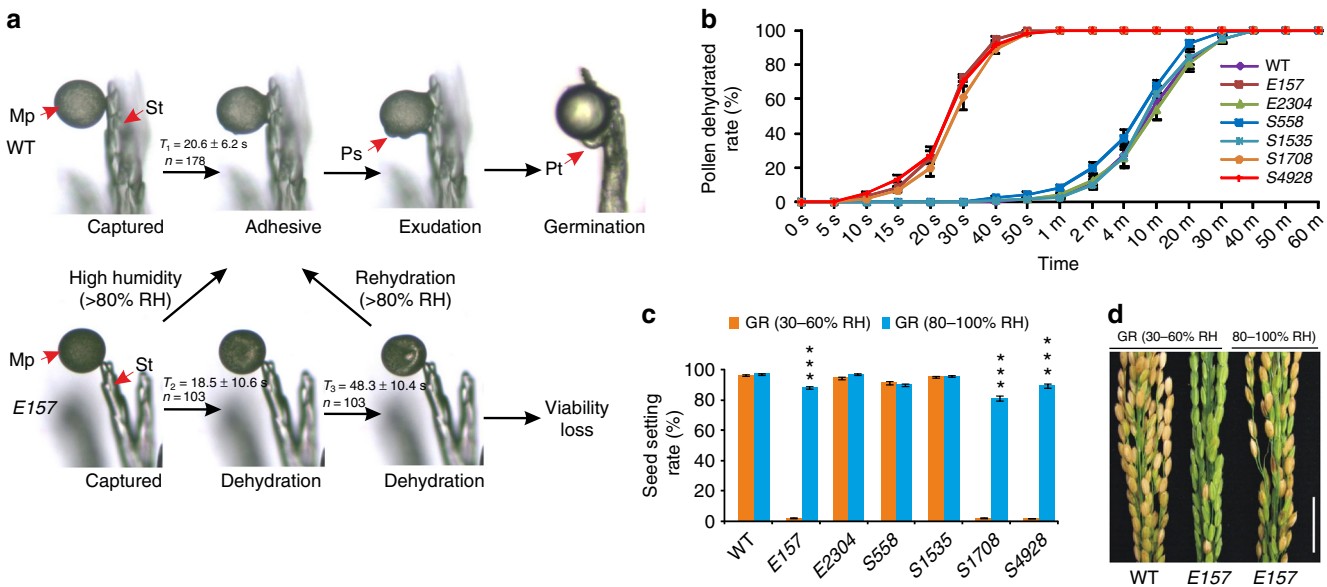

**Fig. 3** Characterization of humidity-sensitive genic male sterility. **a** Germination of WT and mutant pollen on the stigma. $T_2$ and $T_3$ refer to the period between, respectively, the time when the pollen came into contact with the stigma and the onset of pollen dehydration, and from the onset of dehydration and complete dehydration. Mp mature pollen, Ps protrusion, Pt pollen tube, St stigma. **b** Pollen dehydration at 27–32 °C, 30–60% RH. The ratio between the numbers of dehydrated and total pollen grains is presented in the form of mean ± s.d., $n = 3$. **c** Seed set of the WT and mutant plants in different humidity. The data are presented as means ± s.d., $n = 15$. ***$P < 0.001$, Student's $t$ tests. GR growth room. **d** Spikelets of the WT and mutant *E157* plant grown in different humidity. Scale bar, 2 cm

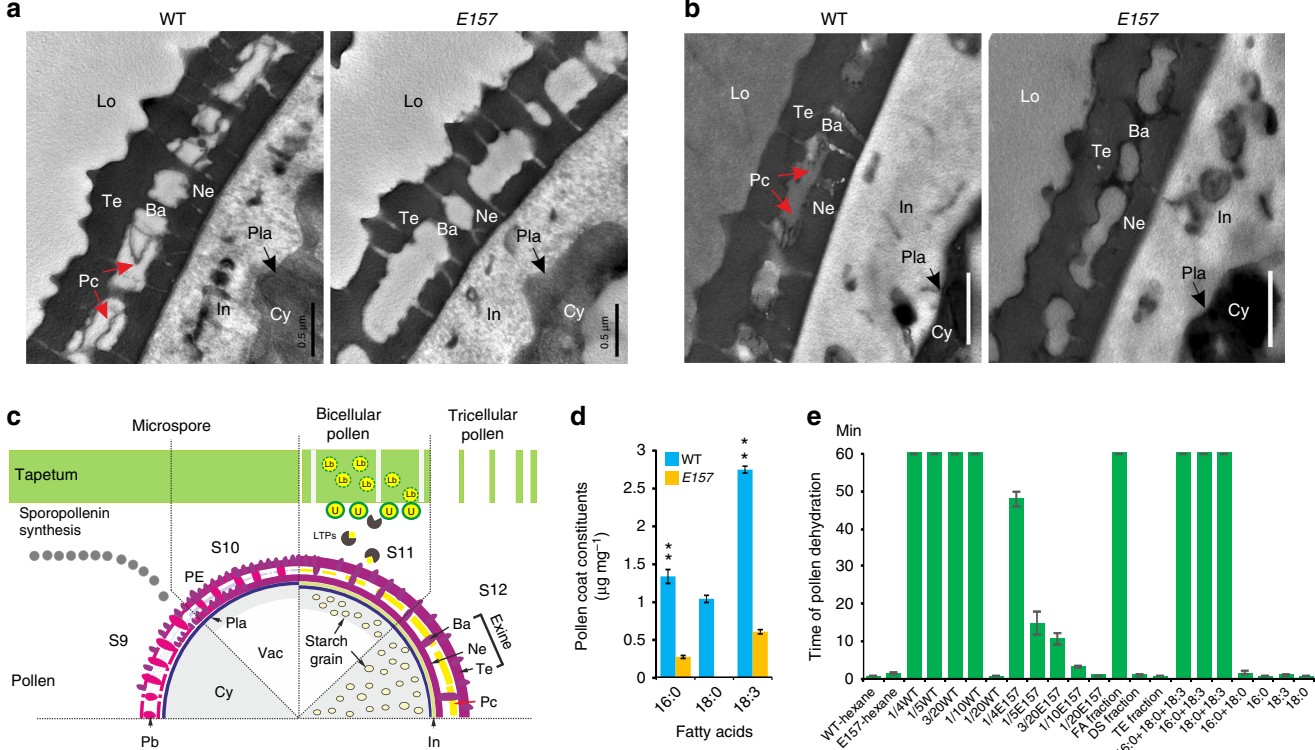

**Fig. 4** Identification of pollen coat chemicals. TEM analysis of ultramicrotome (**a**) and cryo-ultramicrotome (**b**) sections of the WT and *E157* pollen wall at S14. **c** Proposed mode of pollen coat formation during stages S9 to S12. Ba bacula, Cy cytoplasm, In intine, LTPs lipid transfer proteins, Lo locule, Ne nexine, Pc pollen coat (indicated by red arrows), PE primexine, Pla plasma membrane, Te tectum, U ubisch body, Vac vacuole. **d** Analysis of pollen coat lipids of the WT and *E157* plants. The values indicate means ± s.d., $n = 5$, **$P < 0.01$, by Student's $t$ test. **e** Pollen dehydration rates when treated with pollen coat extracts and chemicals. *E157*-hexane: *E157* pollen treated by hexane, WT-hexane: wild-type pollen treated by hexane, abbreviates of fraction and chemical names are listed in Supplementary Table 4. Times of pollen ($n = 100$–150) dehydration are presented in the form of mean ± s.d., $n = 3$. Scale bar, 0.5 μm

the WT PCE (w/w), was not detected in the mutant *E157* pollen coat (Fig. 4d).

Notably, exogenous application of WT PCE prolonged the time to dehydration of the mutant *E157* pollen from 1 to 2 min to more than 1 h when at least 1/10 dilution of WT PCE (20 mg mL$^{-1}$) was added. In contrast, *E157* PCE had no obvious effect (Fig. 4e). Fractionation of the WT PCE demonstrated the three major components separated by silica gel chromatography, fatty acids (FA fraction), dehydrated sterols (DS fraction), and triterpene esters fraction (TE fraction), with the ratios of 38:29:33 (w/w/w) (Supplementary Fig. 10). When exogenously applied, it was the FA fraction that was able to prolong mutant pollen hydration[23]. The FA fraction mainly contains three fatty acids (16:0, 18:0, and 18:3) with ratio of 23:19:49 (w/w/w). Application of single fatty acids was unable to effectively maintain mutant pollen hydration, whereas mixtures of 16:0, 18:0, and 18:3 maintain mutant pollen hydration (concentrations above 20 mg mL$^{-1}$). Further investigation demonstrated that mixtures of only two fatty acids such as 16:0 and 18:3 in a ratio of 23:49 (w/w), or 18:0 and 18:3 with ratio of 19:49 (w/w), are sufficient for prolonging mutant pollen hydration, but the mixture of 16:0 and 18:0 did not produce this effect (Fig. 4e).

**OsOSC12 is a bicyclic triterpene poaceatapetol synthase**. Our GC–MS analysis reveal a minor peak at a retention time of 20.3 min in the anther extract profiles of WT and three 'silent' mutants, that are absent in the three loss-of-function mutants (Fig. 5a). High-resolution electrospray ionization (ESI) mass spectrometry analysis detected a molecular ion peak at an $m/z$ of 427.3929 (M + H$^+$), predicting a molecule of constitution $C_{30}H_{50}O$ (Supplementary Fig. 11). Structural data obtained from nuclear magnetic resonance (NMR) analysis (Supplementary Fig. 11, Supplementary Data 1) of ~1 mg of this compound isolated from 6 kg of rice panicles and mass spectrum (Supplementary Fig. 12) identified this as the bicyclic triterpene poaceatapetol (systematic name polypoda-7,13E,17E,21-tetraene-3β-ol). *Pichia pastoris* yeast cells expressing WT OsOSC12 and proteins from the three non loss-of-function mutants *E2304*, *S558*, and *S1535* synthesized poaceatapetol (Supplementary Fig. 12), while those yeasts expressing proteins of loss-of-function mutants, *E157*, *S1708*, and *S4928* could not produce any of triterpenes. This confirms that OsOSC12 catalyzes the cyclization of 2,3-oxidosqualene into poaceatapetol and is therefore a poaceatapetol synthase (OsPTS1).

Comparative analysis of the non-saponified samples from the WT vs. mutants both of anther (S12) and pollen coat, uncovered three fatty acid (C16:0, oleic acid (C18:1), and C18:0) esters of poaceatapetol in the WT pollen coat (Fig. 5b, c and Supplementary Figs. 13, 14). Further analysis (Supplementary Figs. 13, 14) indicates 0.31 ± 0.01 µg mg$^{-1}$ (mean ± s.d., $n = 5$) triterpene esters rather than free triterpenes are deposited in the pollen coat. However, exogenous application of the TE fraction that includes triterpenes esters could not maintain pollen hydration (Fig. 4e), indicating that poaceatapetol or its derivatives affect pollen dehydration by reducing the accumulation of fatty acids in the pollen coat (Fig. 4c). TEM observation also shows that lipid bodies in the WT tapetum are gradually degraded after S11, but are still maintained at high density and integrity in the *E157* tapetum after this stage (Supplementary Fig. 9). These results suggest that deficiency in poaceatapetol synthesis would hinder the transport of fatty acids and other metabolites from tapetum to pollen coat.

**HGMS-based hybrid production**. As the HGMS lines are male fertile when RH around the time of anthesis stays reliably above 80% but sterile at low RH, we tested whether they could be used for hybrid grain production in places the RH regime is low (<60%). Under field conditions in Beijing in 2012, the seed setting rate on *E157* and *S4928* panicles in the presence of WT pollen was ~30% (Fig. 6a, b). This level of fertility is comparable to what can be achieved using cytoplasmic male sterility (CMS)- or P/TGMS-based methods[24,25]. The proportion of true F$_1$ hybrid seed, as confirmed by sequencing analysis, was >93% ($n = 300$) (Fig. 6a, b). *OsOSC12/OsPTS1* appears to be specific to grass species, and homologs are present in important cereal crops (Fig. 6c). Relevant transcripts, poaceatapetol synthase, and poaceatapetol, were all detected in the anthers of barley, maize, and wheat (Fig. 6d–f, Supplementary Fig. 15).

## Discussion

Plant OSC enzymes are known for the diverse cyclization mechanisms they employ in driving an enormous variety of triterpenoids[20]. Here, OsOSC12 committing a new triterpene pathway producing poaceatapetol and its esters. The three fatty acids, palmitic acid, linolenic acid, and stearic acid, in the pollen coat prevent pollen over dehydration. But it is not clear how does this triterpene pathway control the accumulation of fatty acids in the pollen coat. Poaceatapetol or it derivatives could interact with the membrane of lipid body, controlling the degradation process. We indeed observed that degradation of lipid body is delayed in the mutant tapetal cell comparing to that of the WT. Alternatively, they could act as signal molecules in the regulation of lipids transportation. The function of poaceatapetol and it derivatives in accumulation of fatty acids in the pollen coat will be elucidated through cloning more HGMS mutant genes and analyzing their functions.

Current rice farming practices make heavy use of hybrid varieties. Hybrid varieties using two-line hybrid systems relies on conditional male sterility. Although 427 P/TGMS-based two-line rice hybrid combinations had been registered in China by 2010, two-line hybrid rice accounted for only ~20% of the total area of hybrid rice cultivation[8]. TGMS is very sensitive to temperature, so that any fluctuation outside the range 22–24 °C has a major impact on sterility[3] while PGMS is also influenced by photoperiod requiring day length above 13.5–14 h during development of younger panicles[2]. These strict requirements limit two-line hybrid rice production and cultivation. In contrast, HGMS is not influenced by temperature and photoperiod and so could potentially be used in hybrid rice breeding in areas where the RH is below 60%.

Although unexpected fluctuations in humidity, especially caused by rain during flowering, might impede HGMS-based hybrid seed production there are many suitable dry regions, such as in Urumqi, Xinjiang, China, where RH above 80% was only observed three times between 2005 and 2014 (Supplementary Fig. 16). More extensive field trials in additional locations would be required for further development. Given that the causal gene for the HGMS trait is now known to be *OsPTS1*, back-crossing the trait into any genetic background should be feasible. Likewise, the conservation of PTS1 in grass species suggests HGMS may be applicable to other important crops such as wheat, barley and maize via identification of induced *PTS1* loss-of-function mutants or use targeted knockout strategies.

## Methods

**Plant materials and growing conditions**. Rice plants were grown either in a growth room delivering a 12 h photoperiod under normal (30–60% RH, 25–35 °C) or high humidity (>80% RH, 25–35 °C) conditions, or in the field at the Institute of Botany, Chinese Academy of Sciences, Beijing (40–70% RH, 30–39 °C). The *OsOSC12* mutants were selected from a TILLING population as described below. Selected mutant plants were backcrossed to WT (*Oryza sativa* L. *cv.* Zhonghua11)

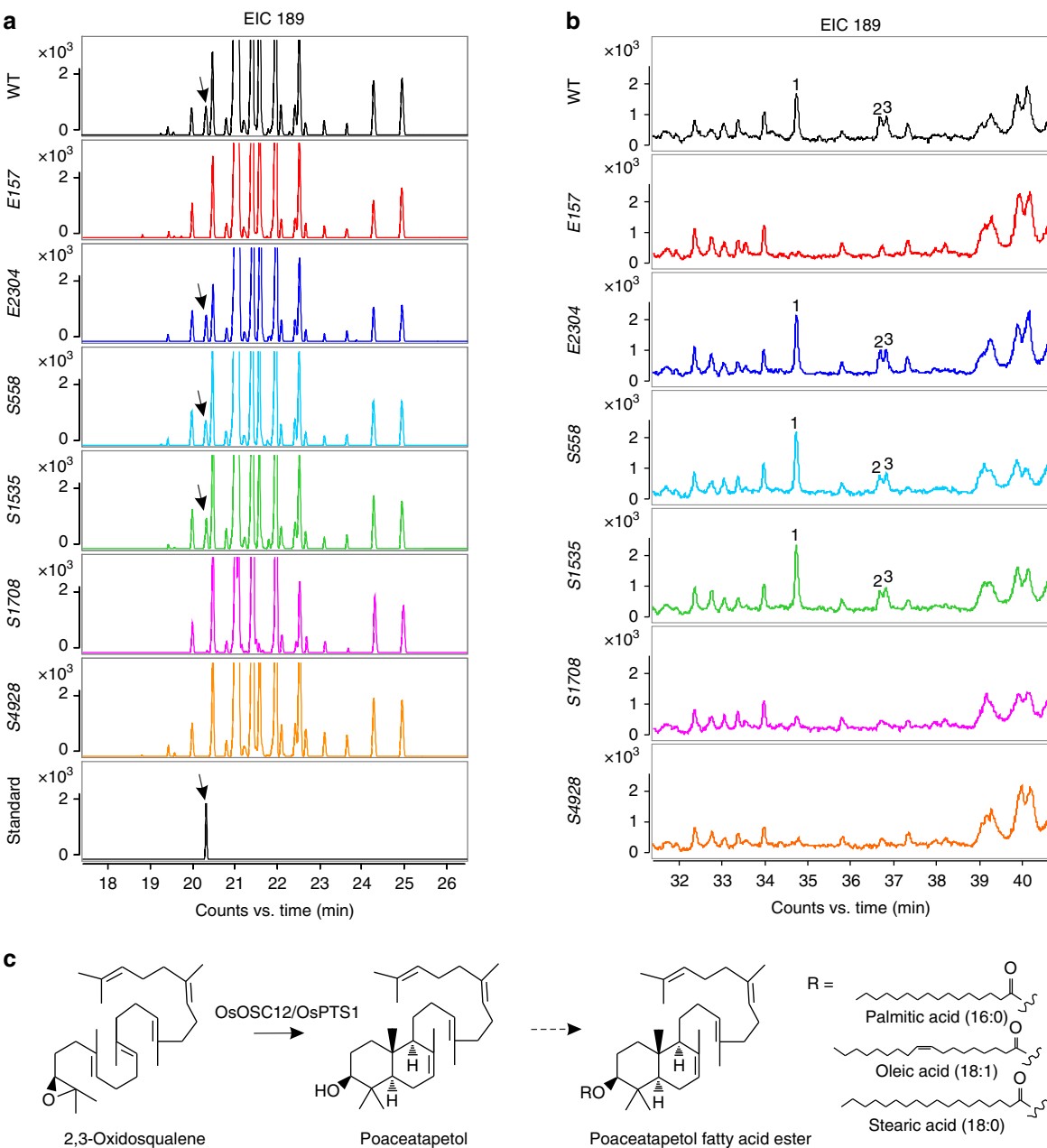

**Fig. 5** The OsOSC12 committed triterpene ester pathway. **a** GC–MS profiles of the triterpene alcohol fraction of anther (S12) extracts from *OsOSC12* mutant and WT plants. The arrow indicates the peaks present in the WT and the "silent" mutants, but absent in the loss-of-function mutants. Standard, purified poaceatapetol. EIC 189, extracted ion chromatograms at *m/z* 189. The arrow indicates the poaceatapetol peak. **b** GC–MS profiles of the triterpene esters of anther (S12) extraction from *OsOSC12* mutant and WT plants. 1: poaceatapetol palmitic acid (16:0) ester; 2: poaceatapetol oleic acid (18:1) ester; 3: poaceatapetol stearic acid (18:0) ester; EIC 189, extracted ion chromatograms at *m/z* 189. **c** The OsOSC12 committed triterpene pathway for biosynthesis of poaceatapetol esters

to remove background mutations. Selected mutant plants from *E2304*, *S558*, *S1535*, *S1708*, and *S4928* were crossed to *E157* for allelism test.

**Northern blotting analysis**. Rice tissues were snap-frozen in liquid nitrogen and ground with a mortar and pestle. Total RNA was extracted from the resulting powder using the TRIzol reagent (Invitrogen, Carlsbad, CA, USA). Gel analysis of the resulting RNA preparation was carried out using standard methods[26]. RNA probe labeling and hybridization were performed at two different stringency levels (55 °C and 68 °C), with a washing protocol as described in the manual (version Jan 2006) provided with the DIG Northern Starter Kit (Roche Diagnostics, Mannheim, Germany).

**Quantitative real-time PCR analysis**. Anthers were excised at various developmental stages, as determined from the length of the anthers and checked using light microscopy. After snap-freezing the material, RNA isolation and cDNA synthesis

was carried out using standard methods. Gene specific primers (OSCA8_1, _2, sequences given in Supplementary Table 3) were designed to amplify 862 bp of the *OsOSC12* sequence. An *Actin* gene (*Os03g0718100*, http://rapdb.dna.affrc.go.jp/) fragment (amplified using primers OsACT_1 and _2, see Supplementary Table 3) was used as the reference sequence. qRT-PCR was carried out using a SYBR Green ER qPCR SuperMix Universal Kit (Invitrogen, Carlsbad, CA, USA), applying an amplification regime of 94 °C for 5 min, followed by 40 cycles of 94 °C for 30 s, 54 °C for 30 s, and 72 °C for 45 s. Experiments included three technical replicates and at least three biological replicates.

**Rice transformation**. The ubiquitin promoter was cloned into the binary vector pCAMBIA1301 by replacing the p35S promoter. The coding sequence of *OsOSC12* (*Os08g0223900*, http://rapdb.dna.affrc.go.jp/) was inserted into the modified pCAMBIA1301 vector under the control of ubiquitin promoter to generate the transformation construct *Ubi*pro::*OsOSC12*. The resulting construct and the control

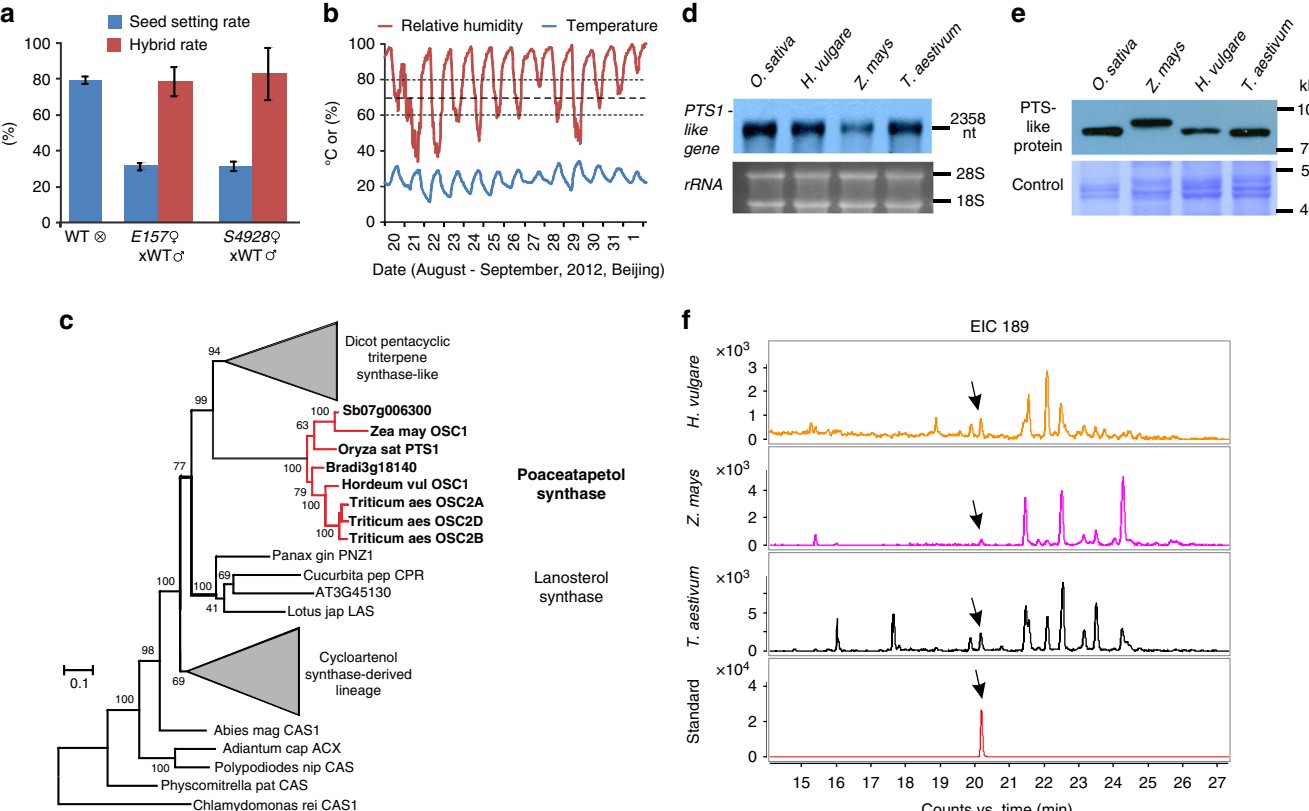

**Fig. 6** HGMS-based hybrid seed production. **a** Seed setting rate and hybrid rate of the crosses of *E157* and *S4928* plants, with the WT plants. The data are presented as mean ± s.d., $n = 15$ panicles for seed setting rate, $n = 300$ seeds for hybrid rate. **b** The variation in RH and temperature at the field site over the period of anthesis. Data were recorded at 1 h intervals by a detector suspended 10 cm above the canopy. **c** A simplified phylogeny of plant OSCs, based on their coding sequence. A maximum likelihood tree was constructed assuming the GTR + Γ + *I* model. The gray triangles indicate groups of OSCs derived from the same lineage. Sequences shown in bold indicate poaceatapetol synthase-like genes. **d** Northern blotting analysis of transcription in the anthers of four major cereal species. **e** Western blotting demonstrates the presence of OsOSC12-like proteins in the anthers of other cereal species. **f** The GC/MS profile of the triterpene alcohol fraction of extracts of barley, maize and wheat anthers. Standard, purified poaceatapetol; EIC 189, extracted ion chromatograms at *m/z* 189. The arrow indicates the poaceatapetol peak

vector pCAMBIA1301 were transformed into *Agrobacterium tumefaciens* strain EHA105 as a means of introducing it into rice calli induced from mature *E157*, *S4928*, *S1708*, and WT embryos. Transgenic calli were selected on a Murashige and Skoog[27] medium containing 50 mg L$^{-1}$ hygromycin B (Roche Diagnostics, Mannheim, Germany). Hygromycin resistant regenerants were transplanted into soil and grown either in a greenhouse or in the field.

**Monoclonal antibody preparation and testing**. A monoclonal anti-OsOSC12 antibody was generated by utilizing a bacterially expressed one of hydrophilic OsOSC12 protein fragment (aa 3–203) as an elicitor (Abmart, Shanghai, China). The specificity of the antibody was tested by immunoblotting of proteins extracted from roots, leaves, and flowers of WT and *OsOSC12* over-expressing rice plants.

**Protein extraction and western blotting analysis**. To extract protein, 200 mg of powdered frozen rice tissues were suspended in 500 µL of Plant Protein Extraction Reagent (CWBIO, Beijing, China), followed by vortexing, incubating at 0 °C for 20 min, centrifuging (7100 × *g*, 20 min, 4 °C), mixing the supernatant with an equal volume of 2× SDS loading buffer and boiling for 10 min. A 5 µL aliquot of each sample was loaded onto an 8% SDS-PAGE gel and electrophorezed at 100 V for 20 min then at 150 V for 50 min. The gels were subsequently incubated in 25 mM Tris, 192 mM glycine, 20% methanol for 20 min, while a PVDF membrane was wetted in methanol (5 min) and then equilibrated for 15 min in the same buffer. A wet transfer cell (Bio-Rad, Hercules, CA, USA) was used to transfer proteins from the gel to the membrane (4 °C, 80 min, 100 V). The membranes were blocked by exposure to 5% w/v powdered skimmed milk in Tris buffered saline with 0.05% (v/v) Tween 20 (TBST pH 7.4, 10 mM Tris, 100 mM NaCl) overnight at 4 °C. A monoclonal rice anti-HSP (heat shock protein, accession Q69QQ6, 80 kDa, Cat. AbM51099-31-PU, BGI, Beijing, China) produced in mice was diluted in 1/10,000 and used as a control. The membranes carrying the transferred proteins were incubated with an anti-OsOSC12 antibody (1/500 dilution in TBST and 1% w/v skimmed milk) for 1 h at room temperature, washed in TBST at room temperature

(3 × 10 min), incubated with a goat anti mouse IgG-HRP conjugated secondary antibody (Cat. CW0102 CWBIO, Beijing, China) diluted 1/5000 in TBST and 1% w/v skimmed milk, and well rinsed. Visualization of the conjugated secondary antibody was performed by supplying SuperSignal West Dura Extended Duration Substrate (Thermo Scientific, Shanghai Division, China) for 5 min, followed by exposure to X-ray film.

**In situ hybridization**. WT anthers at various developmental stages were fixed in FAA (50% v/v ethanol, 5% v/v acetic acid, 3.7% v/v formaldehyde) for 16 h at 4 °C, and dehydrated by passing through an ethanol series (50–60–75–80–95–100–100%). The samples were then passed through a xylene–ethanol series, then embedded in Paraplast (Sigma-Aldrich, St. Louis, MO, USA) with six changes of Paraplast. The blocks were sectioned into 8 µm slices using an RM 2235 (Leica, Wetzlar, Germany) rotary microtome and mounted onto Poly-Prep slides (Sigma-Aldrich, St. Louis, MO, USA). The sections were hybridized to either a sense or an antisense *OsOSC12* probe. Probes were synthesized via PCR: a 273 bp *OsOSC12* cDNA fragment (nucleotides 350–622, counted from the transcription start codon) was amplified from panicle-derived cDNA using primers incorporating the T7 polymerase binding site. For the antisense probe, the primer sequences were ANTISENSE_1, _2, while for the sense probe they were SENSE_1, _2 (Supplementary Table 3). Probes were labeled with digoxigenin (Cat. 11175041910, Roche Diagnostics, Mannheim, Germany) by transcribing the template using T7 polymerase (Roche Diagnostics, Mannheim, Germany), following the manufacturer's instructions. The sections were deparaffinized in xylene, rehydrated through a graded ethanol series and then air-dried. They were treated with 4 µg mL$^{-1}$ proteinase K in 100 mM Tris-HCl, pH 8.0, 50 mM EDTA at 37 °C for 30 min and then washed twice with distilled water for 5 min each. The sections are washed as following: twice with 1× PBS (130 mM NaCl, 7 mM Na$_2$HPO$_4$·7H$_2$O, 3 mM NaH$_2$PO$_4$·H$_2$O) for 2 min, 0.2% glycine in 1× PBS for 2 min, twice with 1× PBS for 2 min. The sections were subsequently treated with 40 mL 1× PBS with 0.5% acetic anhydride and 100 mM triethanolamine for 5 min at room temperature, washed

twice in 1× PBS for 5 min. Sections were incubated at 50 °C for 14–16 h with coverslips in hybridization buffer (100–120 μl per slide) containing the probes (0.02 μg mL$^{-1}$). The hybridization buffer consisted of 50% deionized formamide, 0.3 M NaCl, 10% dextran sulfate, 10 mM Tris-HCl, pH 7.5, 1 mM EDTA, 100 mM DTT and 500 μg mL$^{-1}$ *Escherichia coli* tRNA. After hybridization, successive washing steps were performed as follows: twice in 0.2× SSC, once in RNase A (5 μg mL$^{-1}$) in STE (0.5 M NaCl, 10 mM Tris-HCl, 5 mM EDTA, pH 7.4) at 37 °C for 30 min, twice in STE at 37 °C for 5 min each, and twice in 0.2× SSC at 55 °C for 30 min each. Immunological detection of the hybridized probes was performed ccording to the manufacturer's manual with some modifications. The slides were soaked with 1× TBS (150 mM NaCl, 100 mM Tris-HCl, pH 7.5) and then incubated with 1× blocking reagent (Roche) in TBS for 45 min. They were further incubated with TBST (1% BSA and 0.3% Triton X-100 in TBS) for 45 min, followed by incubation with the diluted antidigoxigenin alkaline phosphatase conjugate (1:1250) in TBST for 2 h. The slides were subsequently washed four times with TBST for 15 min. The sections were rinsed in reaction buffer (100 mM NaCl, 50 mM MgCl$_2$, 100 mM Tris-HCl, pH 9.5), and then covered with reaction buffer contains 0.34 mg mL$^{-1}$ nitroblue tetrazolium salt and 0.175 mg mL$^{-1}$ 5-bromo-4-chloro-3-indolyl phosphate toluidinium salt. After incubation at room temperature for 12 h in the dark, the color reaction was stopped by immersing the slides in TE (10 mM Tris-HCl, 1 mM EDTA), pH 7.5. The sections were observed under microscopy Olympus BX 51.

**Immunolocalization of OsOSC12.** For the purpose of immunolocalization, WT anthers at various developmental stages were fixed in 4% w/v paraformaldehyde in 0.1 M PBS (pH 7.0) for 16 h at 4 °C, dehydrated by passing through an ethanol series, and embedded in Paraplast. Microtome sections (8 μm) were rehydrated in deionised water, and then treated with 3% v/v hydrogen peroxide for 10 min at room temperature. After rinsing three times in TBS, the sections were exposed to 10 mM citrate buffer (pH 6.0) at 95 °C for 10 min, then rinsed three times in TBST and blocked by flooding with 5% w/v skimmed milk in TBST overnight at 4 °C. The sections were incubated with an anti-OsOSC12 antibody (diluted 1/50 in 1% w/v skimmed milk) for 1 h at room temperature. Following three washes in TBST, the sections were incubated with goat anti-mouse IgG-HRP (Cat. CW0102 CWBIO, Beijing, China), diluted 1/200 in 1% w/v skimmed milk, for 1 h at room temperature. Following three washes with TBST, the sections were treated with 3,3′-diaminobenzidine solution before observation by light microscopy.

**Selecting de novo OsOSC12 mutants using the TILLING method.** Grains of cv. Zhonghua11 were obtained from the Chinese Academy of Agricultural Science (Beijing, China). The mutagenesis treatment was carried out in 500 mL flasks, each of which contained 25 g grains; the period of exposure to 2 mM sodium azide (SAZ) was 6 h, while the exposure to 80 mM ethyl methane sulfonate (EMS) was overnight (18 h). The 4950 M$_1$ fertile plants were allowed to self fertilize. DNA was extracted and pooled from the seedling per M$_2$ plant. The DNA pools with each mixed by 8 individual DNA samples were subjected to the TILLING procedure directed at *OsOSC12;* three pairs of gene-specific primers were utilized, one targeting exon 6 to 7 (primers Ex6-7_1, _2), the second exon 8 to 10 (Ex8-10_1, _2), and the third exon 16 to 18 (Ex16-18_1, _2) (Supplementary Table 3). Each 10 μL reaction contained 5 ng pooled DNA, 1× EX Taq buffer (Takara, Dalian, China), 0.2 mM dNTP, 0.16 μM of each primer, and 0.5 U EX Taq polymerase (Takara, Dalian, China). The amplification program comprised an initial denaturation (95 °C for 2 min); eight cycles of 94 °C for 20 s, $T_m$ + 3 °C to $T_m$ − 4 °C decrementing 1 °C per cycle for 30 s, 72 °C for 1 min; then 35 cycles of 94 °C for 20 s, $T_m$ − 5 °C for 30 s, 72 °C for 1 min; 72 °C for 5 min. Post amplification, duplex formation was carried out by imposing an initial denaturation (95 °C/10 min), followed by a 0.3 °C stepwise reduction in temperature from 70 °C to 49 °C with 20 s spent at each temperature. Heteroduplexes were cleaved using a crude celery juice extract and processed following[28]. The products were size-fractionated by denaturing poly-acrylamide gel electrophoresis using a Li-Cor 4300 DNA analyzer (LI-COR, Lincoln, NE, USA). TILLING gels were scanned using GelBuddy and Ecotilling gel analysis software[29]. Putative mutations were validated by sequencing on an ABI 3730XL sequencer (Applied Biosystems, Foster City, CA, USA).

**Assay of pollen germination and dehydration.** Pistils were excised and placed upright on agar (1.5%), then dusted with pollen from a freshly dehiscing anther. The behavior of the pollen was tracked by light microscopy. A humidifier was used to generate high RH conditions. To measure the rate of pollen dehydration, pollen from a freshly dehiscing anther was released onto a dry glass slide and monitored by light microscopy. To examine pollen tube growth, the pistils of the pollinated flowers were fixed in ethanol:acetic acid (3:1) solution for 5 min, transferred into 1 N NaOH, incubated at 60 °C for 30 min and stained with aniline blue (0.1% in 0.1 M K$_3$PO$_4$).

**Extraction and analysis of pollen coat.** For small-scale analysis, pollen coats were extracted by soaking 200 mg pollen in 2 mL of either ethyl ether or chloroform for 1 min, 5 min, 20 min, 60 min and 12 h. These extracts were analyzed using an Agilent 7890/5977A GC–MS device (Agilent Technologies, Santa Rosa, CA, USA) equipped with DB-5HT column (30 m × 250 μm internal diameter, 0.1 μm film)

using electron ionization (EI). The inlet, transfer line, and ion source temperatures were set to, respectively, 340 °C, 280 °C, and 250 °C and the oven profile comprised a 2 min period at 60 °C and 1 min at 220 °C, with the temperature increased at 40 °C min$^{-1}$, followed by 8 min at 340 °C, with the temperature raised at 4 °C min$^{-1}$. The flow rate of helium gas was 2.6 mL min$^{-1}$. Splitless injections (1 μL) were used and mass spectral data between m/z 50 and 800 were acquired. For fatty acid analysis, the PCE were esterified by bis-N,N-(trimethylsilyl) trifluoroacetamide (Sigma-Aldrich) in pyridine for 30 min at 60 °C and quantified using standards as following: palmitic acid, stearic acid, and linolenic acid (Sigma-Aldrich, Shanghai, China). For the large-scale extraction of triterpene esters, 200 g of pollen grains were extracted with 2 × 250 mL ethyl ether and the extracts were combined and dried. The crude extracted product was suspended in hexane and separated by using 20 g silica gel column chromatography with a gradient eluents system including dichloromethane in hexane: 10% for 2 CV (column volume), 20% for 10 CV, and 100% for 2 CV. Each fraction collected in 20 mL test tube was monitored by TLC analysis. To prepare WT and E157 PCE extracts for the exogenous supply experiment, 5 g of WT or mutant pollen were extracted as above, yielding 200 mg WT PCE and 100 mg E157 PCE.

**Exogenous application experiment.** The WT and E157 pollen from a freshly dehiscing anther were soaked for 20 min in different treatments (Supplementary Table 4). The treated pollen grains were then placed onto a dry glass slide and monitored by light microscopy. The dehydration time was recorded when over than 90% of pollen was dried, for up to 1 h. Experiments included three technical replicates and at least three biological replicates.

**Triterpene purification and structural identification.** Metabolites were extracted from 100 mg of lyophilized S12 anther tissue. In brief, homogenized samples was saponified in 10% KOH (w/v) in 80% EtOH (v/v) at 70 °C for 1 h and extracted three times with an equal volume of hexane. Crude extracts were derivatized by treatment with 100 μL N-methyl-N-trimethylsilyl-trifluoroacetamide (MSTFA) at 60 °C for 30 min, and analyzed using same GC–MS equipment. The inlet was changed to 280 °C and the oven profile comprised a 2 min period at 170 °C and 4 min at 290 °C, with the temperature increased at 6 °C min$^{-1}$, followed increasing to 340 °C, with the temperature raised at 10 °C min$^{-1}$. The flow rate of helium gas was 0.8 mL min$^{-1}$. Splitless injections (1 μL) were used and mass spectral data between m/z 50 and 650 were acquired. For the large-scale extraction of triterpenes, about 6 kg of rice panicles at the anther development stages S9–S11 were saponified by using the solvent of 10% w/v potassium hydroxide and 80% v/v ethanol. The n-hexane extraction (52.23 g) was dried and separated by column chromatography using silica gel as adsorbent and 6:1 hexane:ethyl acetate as eluent. The triterpene fractions (9.1 mg) were dried and dissolved in methanol and further purified by reverse-phase HPLC on an Agilent 1200 series LC equipped with a semi-preparative column (Eclipse XDB-C18, 5 μm, 9.4 × 250 mm) using the following profile: 90–100% methanol for 20 min, 100% methanol for 45 min at 2.5 mL min$^{-1}$ flow rate. The semi-preparative fractions were monitored by GC–MS. We found a purified compound (poaceatapetol) at retention time 20.3 min and ca. 1 mg compound was obtained from this fraction. To estimate molecular mass, the purified compound was directly injected into an Agilent 6540 Quadrupole time-of-flight mass spectrometer and positive ESI was applied. The ESI capillary voltage was set to 3.5 kV, the nitrogen gas temperature was set to 320 °C, the drying gas flow rate to 12 L min$^{-1}$, the nebulizer gas pressure to 25 psi, the fragment or voltage to 120 V, the skimmer voltage to 65 V and the capillary voltage to 4 kV. A reference mass correction solution 121.0509 and 922.0098 (Agilent Technologies, Santa Clara, CA) was infused during the analysis to improve accuracy. Mass data (from m/z 50 to 1000) were collected and interpreted using Agilent MassHunter B.06 software. The purified compound was analyzed in an 800 MHz NMR device (Bruker Biospin, Karlsruhe, Germany). $^{1}$H- and $^{13}$C-NMR (800 MHz) and 2D NMR (heteronuclear single quantum coherence (HSQC), heteronuclear multiple-bond correlation (HMBC), rotating frame overhauser effect spectroscopy (ROESY), and homonuclear chemical shift (COSY)) were measured, using C$_6$D$_6$ as an NMR solvent and internal standard.

The HR-ESI-MS displayed a pseudo molecular ion [M + H]$^{+}$ at m/z 427.3929, consistent with the molecular formula of C$_{30}$H$_{50}$O (calcd for C$_{30}$H$_{51}$O, 427.3940). The $^{1}$H-NMR spectrum (800 MHz, C$_6$D$_6$) showed the presence of five vinylic methyls (each 3H, s) at $\delta_H$ (ppm) 1.77, 1.65, 1.63, 1.58, and 1.69. Four olefinic protons were observed at $\delta_H$ (ppm) 5.42 (br s), 5.31(br t), 5.33 (br t), and 5.25 (br t), which, together with the eight olefinic carbons displayed in the $^{13}$C NMR, indicated that four double bonds were present in this compound. This above findings suggested that the purified compound had a bicyclic framework. Inspection of the $^{1}$H–$^{1}$H COSY spectrum led to the establishment of spin–spin coupling systems as depicted in bold line in Supplementary Fig. 11c. In the HMBC spectrum, the cross peaks of Me-23 (Me-24)/C-3 ($\delta_C$ 78.7), C-4, and C-5; Me-25 ($\delta_H$ 0.76, s)/C-1 ($\delta_C$ 37.5), C-5, C-9 ($\delta_C$ 54.4), and C-10; and Me-26 ($\delta_H$ 1.77, s)/C-7 ($\delta_C$ 122.5), C-8 ($\delta_C$ 135.2), and C-9 indicated that the purified compound had a bicyclic scaffold with a methyl and a side chain attached to C-8 and C-9, respectively, and a hydroxy group at C-3. HMBC correlations of Me-27/C-13, C-14, and C-15; Me-28/C-17, C-18, and C-19; and Me-29 (Me-30)/C-21 and C-22 furnished the structure of the side chain. The gross structure of this compound was thus established. The stereochemistry of this compound was determined by the ROESY spectrum in

combination with analysis of coupling constant values of $^1H$ NMR and chemical shift values of $^{13}C$ NMR. It should be noted that H-3 resonated as a doublet of doublet with $J$ values at 11.2 and 2.7 Hz in this compound. The large $J$ value of 11.2 Hz indicated that H-3 was axially α-oriented, and accordingly the hydroxyl group at C-3 was β-oriented. The NOE correlations (Supplementary Fig. 11c) of H-3/H$_3$-23 and H-3/H-5 also supported the above conclusion. The NOE correlations of H-13/H$_2$-15 and H-17/H$_2$-19 indicated that both of the $\Delta^{13}$ and $\Delta^{17}$ double bonds were $E$-configured. The relatively upfield shifted resonances for C-27 ($\delta_C$ 16.3) and C-28 ($\delta_C$ 16.1) in $^{13}C$ NMR also suggested that they were $syn$ to C-12 and C-16, respectively, because of the γ-gauche effect[30], which also supported that both of the $\Delta^{13}$ and $\Delta^{17}$ double bonds were $E$-configured. Therefore, this purified compound was identified as polypoda-7,13E,17E,21-tetraene-3β-ol. A literature search revealed that the purified compound was the same as a bicyclic triterpene that has been isolated from bark of Vietnamese *Cratoxylum cochinchinense*[31] and yeast expression of a mutant of β-amyrin synthase from *Euphorbia tirucalli*[32] previously. The NMR data were assigned with the aid of 2D NMR (Supplementary Fig. 11b), which was consistent with those reported[31,32].

**Physicochemical properties analysis**. For IR spectroscopy, the sample was first dissolved in methanol at 0.05 g mL$^{-1}$. A drop of this solution is deposited on surface of KBr cell. The solution was then evaporated to dryness and the film formed on the cell is analyzed directly on a Nicolet IN 10 Micro FTIR spectrometer by transmission mode (Supplementary Data 2). For UV spectroscopy, the same solution was analyzed by the DAD detector of an Agilent 1200 series LC and scanned from 210 to 600 nm (Supplementary Data 2). For optical rotation, the same solution was analyzed on a Anton Paar MCP100 polarimeter and the specific rotation was $[\alpha]^{25}_D = +7.25$ ($c = 0.035$, CHCl$_3$). Values represent means ± s.d. of three technical replicates.

**Heterologous expression in *Pichia pastoris***. Total RNA isolated from rice anther was converted into cDNA using SuperScript II reverse transcriptase (Invitrogen, Carlsbad, CA, USA) following the manufacturer's protocol. The resulting cDNA was used as a template to isolate *OsOSC12* by PCR, using the primer pair OSC8S/OSC8A (Supplementary Table 3) and Phusion DNA polymerase (New England Biolabs, NEB). The PCR program comprised an initial denaturation (98 °C/30 s), followed by 35 cycles of 98 °C for 10 s, 60 °C for 30 s, 72 °C for 45 s, with a final extension step of 72 °C for 7 min. The amplicon was separated by agarose gel electrophoresis and gel-purified using a Gel Extraction Kit (Qiagen, Valencia, CA, USA). The 2.3 kb product was inserted into the pGEM-T-Easy vector (Promega, Madison, WI, USA) for sequencing. *SnOsOSC12* (Supplementary Data 3) was made by synthesis, with yeast-preferred codons incorporated into the sequence[33], then introduced into the pUC57 vector (Shanghai Sangon Biological, Shanghai, China). All mutants were generated by PCR (primers Exxx or Sxxx_Mu_1, _2 in Supplementary Table 3) based site-directed mutagenesis as the manual of QuickChange site-directed mutagenesis kit (Stratagene Inc., La Jolla, CA). The plasmids were double-digested with *Sac* II and *Spe* I and ligated into the yeast expression vector pPICZa A (Invitrogen, Carlsbad, CA, USA) driven by a methanol-inducible promoter. The plasmids were introduced into *P. pastoris* WT strain X33 using electroporation. X33s harboring *SnOsOSC12* and its mutagenized genes were grown at 30 °C in minimal glycerol medium (MGY, 1.34% yeast nitrogen base, 1% glycerol, $4 \times 10^{-5}$% biotin) to OD$_{600}$ = 2–6. The cells were collected by centrifugation, resuspended in minimal methanol medium (MM, 1.34% yeast nitrogen base, $4 \times 10^{-5}$% biotin, 0.5% methanol) to OD$_{600}$ = 1.0 and incubated at 30 °C for 24 h. Cells were collected from 25 mL culture disrupted with 2 mL 10% KOH (w/v) in 80% EtOH (v/v). The refluxed products were extracted twice with 2 mL hexane, and combined both hexane solutions to obtain the crude extract. The extracts were either directly derivatized using MSTFA and analyzed by GC–MS as described above.

**Scanning electron microscopy**. S12 anthers and stigmas sampled simultaneously were fixed in FAA for 24 h at 4 °C, dehydrated by passing through an ethanol series, then through an ethanol–isoamyl acetate series, and finally dried at liquid CO$_2$ critical point for 4 h. Individual anthers and stigmas were mounted on SEM stubs, coated with gold and scanned using a Hitachi S-4800 device (Hitachi, Tokyo, Japan), with the acceleration voltage set to 10 kV. To avoid interference from organic solvents, the S12 anthers were also dehydrated at room temperature, then directly coated with gold.

**Transmission electron microscopy**. WT and mutant pollen grains at stage 14 were fixed in 4% paraformaldehyde and 3% v/v glutaraldehyde in 0.1 M PBS (pH 7.0), then post fixed in 1% w/v OsO$_4$ in PBS. The samples were dehydrated by passing through an ethanol series, then embedded in Spurr's resin (Sigma-Aldrich, St. Louis, MO, USA) which was subsequently polymerized at 60 °C for 24 h. Ultrathin sections (70 nm) were made using a diamond knife microtome (Leica Ultracut R). The sections were placed on 100 mesh copper grids and sequentially stained first in 2% w/v uranyl acetate for 30 min and then in 0.2% w/v aqueous lead citrate for 5 min. Sections were examined using a JEM-1230 device (JEOL, Japan) operating at 80 kV. For cryosectioning, S14 pollen was suspended in 20% w/v BSA in 0.1 M PBS, then transferred into 0.05 mm deep wells, in which they were frozen in a high pressure freezing machine HPF Compact 01 (M. Wohlwend GmbH,

Switzerland). The specimens were then placed in freeze-substitution solution (1% w/v OsO$_4$, 0.1% w/v uranyl acetate in acetone) and transferred into a Leica AFS2 freeze-substitution machine. The freeze-substitution procedure began with holding the samples at −90 °C for 30 h, followed by −60 °C for 24 h, −30 °C for 18 h and −20 °C for 2 h. During the temperature transitions, the temperature was increased gradually (over 8 h for the first transition, 4 h for the second and 1 h for the third). After warming to 0 °C, the specimens were washed three times in acetone and then warmed to room temperature, before infiltrating with SPI-PON 812 resin. The resin was allowed to polymerize at 45 °C for 18 h, followed by 24 h at 60 °C. Ultrathin sections were cut and observed as described above.

**Sequence alignment and phylogenetic analysis**. A multiple alignment of OSC polypeptide sequences was performed, and a codon matrix produced using the Muscle alignment package provided in MEGA v5.1 software[34]. The same package was also used to conduct a phylogenetic analysis, based on codon alignments and the maximum likelihood method, assuming the GTR + Γ + I substitution model.

**Data availability**. The authors declare that all data supporting the findings of this study are provided in the manuscript and its supplementary files or are available from the corresponding author upon request.

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

## Acknowledgements

This work was supported by funds from the National Natural Science Foundation (grant no. 31530050, 31370337 and 31170278), National Transgenic Megaproject of China (grant no. 2016ZX08009) and the Ministry of Science and Technology (grant no. 2013CB127000) of China. We thank C.M. Liu at the Institute of Botany of the Chinese Academy of Sciences (IBCAS) for providing the rice EMS population. We thank F.Q. Dong and Y.H. Xiao at IBCAS for technical assistance with TEM and SEM analyses. We thank W.Z. He at the National Institute of Biological Sciences, Beijing (NIBS) for technical assistance of analysis of cyro-ultramicrotome sections. We thank Y.N. Wang at Capital Medical University and N. Xu at Tsinghua University for NMR analysis. We thank Z. Xue, B. Han (IBCAS) and L.D. Han (Chinese academy of agricultural sciences) for assistance in metabolite analysis. We thank A. Osbourn, R. Thimmappa, H.W. Nutzmann and A. Orme at John Innes Centre, H. Ma at Fudan University and R. Peters at Iowa State University for the critical comments.

## Author contributions

Z.X. and X.Q. designed all experiments. Z.X., X.X., Y. Zhou., Y. Zhang., D.L. and B.Z. performed the experiments. Z.X., X.X., X.W., L.D. and X.Q. contributed to data analysis and manuscript preparation. Z.X., X.W. and X.Q. wrote the manuscript.

## Additional information

**Competing interests:** The authors declare no competing financial interests.

