## [Peer Review File · Nature Communications]

Reviewers' comments:

Reviewer #1 (Remarks to the Author):

Xue reported a mutation of rice a triterpene biosynthetic pathway gene OsOSC12/OsPTS1 causing in humidity-1 sensitive genic male sterility (HGMS) in rice. The authors conducted extensive investigation on the humidity associated fertility of the mutant and they concluded that this mutant is male sterility at low relative humidity (RH < 60%), and is fully male fertile at high RH (> 80%). The HGMS trait is novel for hybrid breeding compared the current used P/TGMS in two line hybrid rice system. Furthermore the sequence and expression pattern of OsOSC12/OsPTS1 are conserved in grass plants, suggesting the evolutionary significance of the OsOSC12/OsPTS1 pathway.

Comments:

1) Line 34-36, Line 115-117, the authors mentioned the lipidic body (Supp Fig 6), however, this observation is incomplete. I suggest the authors to observe the changes of tapetal cells, and lipidic structures called Ubich Body attached in the inner surface of tapetal cells by referring the literature by Shi et al., *The Plant Cell*, 2011, 23: 2225-46; Li et al., *The Plant Cell*, 2010, 22(1): 173-190 at various stages from stage 8 to 11.

2) Figure 1 shows the expression of OsOSC12 in tapetal cells. The expression signal is hard to see in which cell type, I suggest the authors to show one anther lobe instead of four and see the detailed expression of OsOSC12.

3) Supplemental file and line 77 in the main text mentioned the method for extracting pollen coat (Supp Fig 7), I am not sure this method can only extract the chemicals of pollen coat? It has been reported that rice pollen wall may share common pathway with that wax and cutin in the anther (. Zhang, DB., Yang XJ and Shi JX. (2016). "Role of lipids in plant pollen development". In *Lipids in Plant and Algae Development, Subcellular Biochemistry* 86, pages 315-337. Editor: Nakamura, Yuki, Li-Beisson, Yonghua. Springer Science+Business Media New York; Zhang, DB., and Li H. (2014). "Exine Export in Pollen". In *Plant ABC Transporters, Signaling and Communication in Plants*, 22, pages 49-62. Editor: Markus Geisler. Springer Science+Business Media New York.)

4): The statement on Line 117-119: "These results indicate that poaceatapelol is involved in regulating the transportation of fatty acids and other metabolites from tapetum to pollen coat by a novel regulatory system." has no direct evidence.

5): Figure 4C: the model about the pollen coat and pollen wall is too speculative and there is no chemical labelling and supportive data. Please refer to the model by Li and Zhang. *Plant Signaling & Behavior*, 2010, 5(9): 1121-1123.

6) For the rice PGMS lines, please mentioned the csa line (PNAS, 2013 110 (1) 76-81)

Reviewer #2 (Remarks to the Author):

The manuscript by Xue et al. « Deficiency of a triterpene pathway results in humidity-sensitive genic male sterility in rice » presents a functional characterization of an oxidosqualene cyclase from rice named OsOSC12 that is one of the twelve OSCs reported in rice. OSC12 displays a very strong tapetum-specific expression at stages where microspores are formed. This is nicely shown by in situ nucleic acid hybridization and immunolocalization. The expression pattern of the 11 other OSCs is not documented using the same approaches as far as I know. The authors have performed a tilling experiment that resulted in the isolation of an allelic series in OSC12. Three mutations (E157, stop255 ; S1708, G477E ; S1535, E747K) define strong alleles implied in a very interesting phenotype of conditional male sterility at low relative humidity. A set of very clear experimental data showed that mutant pollen grains dehydrate on stigmas, and this is due to a defect in the structure or composition of the lipidic pollen coat. In fact, the exogenous application of mixtures of fatty acids to the mutant pollen prevented dehydration. This is reminiscent of the results published by Wolters-Art et al (*Lipids*

are required for directional pollen-tube growth. Wolters-Arts M, Lush WM, Mariani C. Nature. 1998 Apr 23;392 (6678):818-2) showing the role of lipids in controlling hydration. The authors propose that compounds lacking in the strong alleles are palmitate, stearate and oleate derivatives of a bicyclic triterpene converted from oxidosqualene by OSC12. This triterpene is identified as polypoda-7,13,17,21-tetraene-3-beta-ol also known as poaceatapelol. The analysis of pollen coat chemicals (Figure 4) indicates a strong decrease of fatty acids in extracts from the mutant E157 compared with those from the wild-type (about 5 micrograms per mg fatty acids in the wild-type and less than one in the mutant). It should be important to quantify also the amount of triterpene poaceatapelol in the pollen coat chemicals, to see if there is an equivalent decrease. In addition, the *Pichia pastoris* assay showing the production of poaceatapelol upon expression of WT OSC12 should be implemented to prove the loss of function of E157, S1708, S1535). Apparently, the lack of poaceatapelol synthesis in the anthers is disrupting the accumulation of lipids in the pollen coat and this has a major deleterious effect on pollen hydration in low relative humidity at the onset of pollination. I fear that the characterization of the chemical phenotype of WT and mutant pollen suffers from a lack of explanations : the GC-MS profile of poaceatapelol esters is based on an extracted ion chromatogram at m/z 189 as it is the case for poaceatapelol from a saponified fraction (containing free triterpenes), whereas the mass spectra of the poaceatapelol esters of C16 :0, C18 :0 and C18 :1 does not show such an ion (Supp. Figure 10). Additional comments are listed below regarding supplementary figures and tables.

Typos (some).

Line 91, 112. fraction, not Fraction.

Line 104. polypoda-7,13,17,21-tetraene-3-beta-ol (not polypoda-7,13,17,21-tetraene).

Line 114. Affect (not effect).

Line 272. Triterpene esters (not ester).

Supplementary file.

Supplementary Figures and Tables

Line 290.

Supplementary Figure 4. Are the seed set data shown in a panel (b) recorded at 30-60% RH ?

Line 302.

Supplementary Figure 7. These pieces of data correspond to the chemical analysis of pollen coat described line 139-143 (extraction of 200g of pollen grains).

I believe there is a concern with the GC diagrams (TIC or EIC) shown in panel b. In fact, the GC runs last 47 minutes as indicated on the horizontal axis of the chromatogram. The oven temperature program (lines 132-135) indicates that the peak framed as a triterpene ester at 39min is eluted at 340°C (at the start of the last 8' plateau of 340°C). It would be interesting to see the scan m/z 50-800 of that peak by zooming in (whole mass spectrum of the peak), not only the EIC at m/z 189. In this experiment if the peak at 39 minutes is a triterpene ester it is not clear to me whether it could be a C16:0, or C18:0, or C18:1 ester of poaceatapelol. In addition, what is the peak at 27 minutes present in WT and E157 on the TICs at EICs m/z 189 ? Also, some peaks between 20-22 minutes in the mutant E157 are increased (EIC m/z 189). What are their mass spectra ?

Line 326.

Supplementary Figure 8.

This shows the structural elucidation of the bicyclic triterpene of rice pollen coat as poaceatapelol. Line 329 this compound should be written « polypoda-7,13,17,21-tetraene-3-beta-ol », not polypoda-7,13,17,21-tetraene. In (a), a compound is collected by semi-preparative LC for a further NMR

analysis. Would it be possible to have the NMR spectrum as a supplementary file ? In (b), the authors mentioned a reference standard of poaceatapelol. What is the origin of the standard compound ? In (c), the mass spectra data (scan m/z 50-700) should be presented, there is a peak on the TIC of SnOsOCS12 expressed in Pichia, it would have been easy to zoom in. In (d), the referee is confused with the whole data set. In fact, if the mass spectrum shown in (d) corresponds to the peak produced in Pichia as shown in (c) after extraction of the triterpene and GC-MS analysis of TMS-derivatives, how is the fragmentation explained with respect to the ion m/z=189 ? Please clarify the mass spectral data. The fragmentation of polypoda-7,13,17,21-tetraene-3-beta-ol as a TMS derivative should be drawn.

Line 337.

Supplementary Figure 9.

Please specify the type of GC-MS analysis that is performed. It is important to figure out what is the m/z 189 ion to understand that poaceatapelol is present as a fatty acid ester in the pollen coat.

Line 340.

Supplementary Figure 10.

The authors should indicate if this is a zoom of the TICs shown in Supplemental Figure 7. What is the ionization energy employed to generate ions shown here ? What is the link with m/z 189 ?

Methods

Line 3. Growth room conditions. It is not written whether RH is controlled or not in growth room. The authors propose that 30-60% RH is normal conditions but in my opinion 30% is a value that should be defined as dry. What is the device that enables RH measurement in growth rooms ?

Line 7. Rice TILLING mutants. Selected mutants were backcrossed to WT cv Zhonghua 11. How many backcrosses ?

Three BCs should clean 87.5% of the background, in principle.

Line 22. qRT-PCR analysis of OsOSC12. The size of the amplicon obtained after qRT-PCR should be given (instead of « a portion »). Please indicate the OsGENE number for the Actin reference.

Line 40. Anti-OSC12 antibody. What is the reason for choosing the 3-203 protein fragment ?

Lines 100-103. Tilling experiment. The size of the mutagenesis experiment as number of M1 and size of pooled M2 should be given.

Line 130. Analysis of pollen coat. Is it a high temperature GC-MS protocol that is described here ?

Line 137. Fatty acid analysis of pollen coat extracts (PCE) obtained (line 128) by soaking pollen in organic solvents.

Does it mean that pollen coat extracts described in lines 128-130 contain free fatty acids ?

Line 154. Although there is a reference 21 in this sentence I think it is important to recall to the reader the type of solvent that was employed to prepare an extract here.

Line 156. The GC column has a size of 29,5 meters. Was it a 30-meter column shortened for normal maintenance of the device ? If this is the case I think it better to write 30-meter according to the manufacturer descriptions.

Lines 161-165. An analysis of triterpenes in the non-saponified and the saponified extracts suggest

that triterpenes are present in the same biological fractions as 3-hydroxy- free compounds and as conjugates of fatty acids that are hydrolyzed in alkaline conditions. If triterpenes are analyzed as TMS - derivatives with MSTFA, why should the GC conditions be different ? This is misleading.

Lines 165-180. Large scale extraction and analysis.

The non saponifiable and fractionated hexane extract of S9-S11 panicles (6 kg) was purified on an Agilent 1200 LC semi-preparative instrument and fractions were monitored by GC-MS to look for a triterpene of interest. What is the fraction that gave a single peak in GC -MS ? There should be one since the authors state (line 172-173) that « the purified compound was directly introduced in the mass spectrometer for M+ estimation, here the spectrometer being an Agilent Q-TOF 6540.

In addition, mass data collected from m/z 50-1000 could have shown the expected M+ for the fatty esters of triterpene (eg M+ 426 (C₃₀H₅₀O, triterpene), + M+280 (linoleate) = 706).

In this case I don't understand why the extract was saponified at the beginning of the whole analytical process, because the exact nature of triterpene esters is then lost.

Line 201. *Pichia pastoris* expression. What is the peculiarity of the strain X33 ? is it a wild-type strain ?

Line 221. Please write « ...were frozen in a high pressure freezing machine HPF Compact 01 (M. Wohlwend »

Typos.

Line 167 : chromatography

Line 202 : derivatization

Line 203 : previously

Line 310, 311 : esters, not ester

Reviewer #3 (Remarks to the Author):

The following comments regard the chemical characterization of the pollen extracts and the triterpenes.

Based on these observations, the manuscript cannot be recommended for publication.

Line 82

At levels of 3.4 and 6.9 %, is the precision of the fold decrease values (4.93 and 4.51) statistically significant?

Lines 98 ff

Are the GC methods used in Supplement Figure 7 and Fig. 5 different? The WT EIC189 chromatogram in the former show baseline/noise between 23 and 26.5 min, whereas the peak of the target cpd and others falls into this retention time window. The peaks in the 38 to 42 minute window in Supplement Figure 7 seem to be congruent with those in Fig. 5 (c), but the baseline/noise level is different.

The isolation of 1 mg of polyoda-7, 13, 17, 21-tetraene from 6 kg of rice panicles is a major undertaking, but not described in any detail.

Figure 5

The structures of the poaceatpetols show a wedged bond for the methyl group attached to an olefinic carbon. This is incorrect as olefinic carbons are not chiral.

Supplement Figure 7

How was the quantification in (a) done? BY total GC-MS peak area? If so, (a) and (b) should be reversed. This also needs to be clarified.

Which solvent was used to generate the extracts in (b)?

No information is provided on the fractionation shown in (c); column material, conditions, yields, etc.

“Star” should be “asterisk”

English language of the figure caption needs a major revision

What does “Yellow square shows the difference” mean?

Supplement Figure 8

The identification of the compound is solely based on ESI HR-MS and comparison of ¹³C NMR data reported in the literature. The typical 1D and 2D NMR data sets and NMR interpretations are not provided. Physicochemical characterization (UV, IR, alpha etc) are also missing.

Regarding the ¹³C based identification of the compound: where was the reference standard obtained from? Or is this a case of confusion between a “reference standard” and a “literature reference”?

Assuming the authors intended to refer to a “literature reference”, the entire identification of the compound hinges on the quality of the assignments made in the 1998 Nguyen et al. publication. While it is likely that the compound isolated by both groups is the same (with the notable potential exception of the absolute configuration), due to the close congruence of the ¹³C NMR data, the validity of the structural assignment made by Nguyen et al. should at least be reviewed critically. Preferably, and reflecting the importance of the isolated compound for the present study, the authors should deduce the structure ab initio, rather than assuming that the published structure is correct.

The data published by Nguyen et al. has largely incomplete ¹H NMR data and interpretations and lacks ²D NMR data completely. Moreover, Nguyen et al. provide no evidence about the absolute configuration and speculate that their compound “belongs to the normal series”. Using this as reference for establishing a structure is not acceptable.

Importantly, for reproducibility, the authors should provide a complete set of spectroscopic and physicochemical data, as is standard in all chemistry and natural product journals.

The comparison of the ¹³C NMR data in (b) does not consider the different decimal precision (one or no decimal place vs. two decimal places). The Nguyen et al. reference lists all values with one decimal place.

The intended word in the a/b/c footnote in (b) is probably “interchangeable”, and “assignment” should be pluralized.

The EI MS shown in (d) is missing interpretation. The description of the analyzed materials

("trimethylsilyl derivative from the yeast extraction expressing a 334 synthetic OsOSC12 with the codon optimization") is unintelligible.

Linguistic Comments

While generally acceptable, English usage throughout the manuscript interferes unfavorably with scientific meaning. This happens in a relatively subtle way, but is important and requires careful revision. To give just one example, the second last sentence in the abstract is incomprehensible, mainly due to the word 'sufficient' (necessary? essential?).

Responds to the reviewers' comments (in blue font):

Reviewer #1 (Remarks to the Author):

Xue reported a mutation of rice a triterpene biosynthetic pathway gene OsOSC12/OsPTS1 causing in humidity-1 sensitive genic male sterility (HGMS) in rice. The authors conducted extensive investigation on the humidity associated fertility of the mutant and they concluded that this mutant is male sterility at low relative humidity (RH < 60%), and is fully male fertile at high RH (> 80%). The HGMS trait is novel for hybrid breeding compared the current used P/TGMS in two line hybrid rice system. Furthermore the sequence and expression pattern of OsOSC12/OsPTS1 are conserved in grass plants, suggesting the evolutionary significance of the OsOSC12/OsPTS1 pathway.

Comments:

1) Line 34-36, Line 115-117, the authors mentioned the lipidic body (Supp Fig 6), however, this observation is incomplete. I suggest the authors to observe the changes of tapetal cells, and lipidic structures called Ubich Body attached in the inner surface of tapetal cells by referring the literature by Shi et al., *The Plant Cell*, 2011, 23: 2225-46; Li et al., *The Plant Cell*, 2010, 22(1): 173-190 at various stages from stage 8 to 11.

Response:

Thanks for the suggestion. More TEM observation of tapetal cell and pollen wall from stage 8 and 12 were added in the revised supplementary Figs. 6 and 7. There is delay of lipid body degradation in the *E157* mutant comparison with the WT. But we did not observe any difference in Ubisch Body between WT and *E157* mutant.

2) Figure 1 shows the expression of OsOSC12 in tapetal cells. The expression signal is hard to see in which cell type, I suggest the authors to show one anther lobe instead of four and see the detailed expression of OsOSC12.

Response:

Yes, Figure 1e has been zoomed in one anther lobe.

3) Supplemental file and line 77 in the main text mentioned the method for extracting pollen coat (Supp Fig 7), I am not sure this method can only extract the chemicals of pollen coat? It has been reported that rice pollen wall may share common pathway with that wax and cutin in the anther (Zhang, DB., Yang XJ and Shi JX. (2016). "Role of lipids in plant pollen development". In *Lipids in Plant and Algae Development, Subcellular Biochemistry* 86, pages 315-337. Editor: Nakamura, Yuki, Li-Beisson, Yonghua. Springer Science+Business Media New York; Zhang, DB., and Li H. (2014). "Exine Export in Pollen". In *Plant ABC Transporters, Signaling and Communication in Plants*, 22, pages 49-62. Editor: Markus Geisler. Springer Science+Business Media New York.)

Response:

Yes, your concern that the rice pollen wall material known as sporopollenin has similar chemical compositions such as the long chain fatty acids as that of pollen coat is right. We were aware of this problem and it has been considered when we designed the extraction method. Previous study found that sporopollenin is not dissolvable in water and organic solvent (Quilichini et al. 2015 *Phytochemistry* 113:170). However, the pollen coat material is very easy to be removed by treatment with organic solvents (Murphy 2006 *Protoplasma* 228:31). We designed a special method for extraction of pollen coat material. Briefly, the intact pollen grains were treated with organic solvents with different polarity for different periods of time, and the broken pollen grains were used as the control. The extractions were monitored by GC-MS analysis and indicated that gentle treatment with the weak polarity organic solvents (such as ethyl ether) or chloroform by less than <5 min can mainly extract pollen coat materials rather than pollen wall chemicals. The method for specifically extracting pollen coat was described in detail in the supplementary method (line 128-130, and Supplementary Figure 8) in the revised manuscript.

4): The statement on Line 117-119: “These results indicate that poaceatapelol is involved in regulating the transportation of fatty acids and other metabolites from tapetum to pollen coat by a novel regulatory system.” has no direct evidence.

Response:

Yes, our results could not directly support this statement, which was deleted in revised manuscript.

5) Figure 4C: the model about the pollen coat and pollen wall is too speculative and there is no chemical labelling and supportive data. Please refer to the model by Li and Zhang. *Plant Signaling & Behavior*, 2010, 5(9): 1121-1123.

Response:

Yes, the model of pollen coat and pollen wall has been revised by referring the literatures (Li and Zhang 2010 *Plant Signaling & Behavior* 5: 1121).

6) For the rice PGMS lines, please mentioned the *csa* line (PNAS, 2013 110 (1) 76-81)

Response:

Yes, the *csa* paper was cited in the revised version of the manuscript (line 128).

Reviewer #2 (Remarks to the Author):

The manuscript by Xue et al. « Deficiency of a triterpene pathway results in humidity-sensitive genic male sterility in rice » presents a functional characterization of an oxidosqualene cyclase from rice named OsOSC12 that is one of the twelve OSCs reported in rice. OSC12 displays a very strong tapetum-specific expression at stages where microspores are formed. This is nicely shown by in situ nucleic acid hybridization and immunolocalization. The expression pattern of the 11 other OSCs is not documented using the same approaches as far as I know. The authors have performed a tilling experiment that resulted in the isolation of an allelic series in OSC12. Three mutations (E157, stop255 ; S1708, G477E ; S1535, E747K) define strong alleles implied in a very interesting phenotype of conditional male sterility at low relative humidity. A set of very clear experimental data showed that mutant pollen grains dehydrate on stigmas, and this is due to a defect in the structure or composition of the lipidic pollen coat. In fact, the exogenous application of mixtures of fatty acids to the mutant pollen prevented dehydration. This is reminiscent of the results published by Wolters-Art et al (Lipids are required for directional pollen-tube growth. Wolters-Arts M, Lush WM, Mariani C. Nature. 1998 Apr 23;392 (6678):818-2) showing the role of lipids in controlling hydration. The authors propose that compounds lacking in the strong alleles are palmitate, stearate and oleate derivatives of a bicyclic triterpene converted from oxidosqualene by OSC12. This triterpene is identified as polypoda-7,13,17,21-tetraene-3-beta-ol also known as poaceatapelol. The analysis of pollen coat chemicals (Figure 4) indicates a strong decrease of fatty acids in extracts from the mutant E157 compared with those from the wild-type (about 5 micrograms per mg fatty acids in the wild-type and less than one in the mutant). It should be important to quantify also the amount of triterpene poaceatapelol in the pollen coat chemicals, to see if there is an equivalent decrease.

Response:

Yes, we quantified the amount of poaceatapelol in the pollen coat. The poaceatapelol was detected in saponified the wild-type pollen at about 0.31 micrograms per mg concentration (Line 114, Supplementary Figure 10) and undetectable in the unsaponified WT pollen, whereas it was undetectable in both the saponified and unsaponified mutant *E157* pollen. This indicated that poaceatapelol was converted into the palmitate (C16:0), stearate (C18:0) and oleate (C18:1) derivatives and accumulated in the wild-type pollen coat (Supplementary Figure 11). Yes, the decrease of major fatty acids (C16:0, C18:0, C18:3) in mutant *E157* pollen coat was associated with the decrease of poaceatapelol (derivatives).

In addition, the *Pichia pastoris* assay showing the production of poaceatapelol upon expression of WT OSC12 should be implemented to prove the loss of function of E157, S1708, S1535).

Response:

Yes, the heterologous expression in *Pichia pastoris* of the WT and its mutants OsOSC12 proteins has been implemented in Supplementary Figure 9c.

Apparently, the lack of poaceatapel synthesis in the anthers is disrupting the accumulation of lipids in the pollen coat and this has a major deleterious effect on pollen hydration in low relative humidity at the onset of pollination. I fear that the characterization of the chemical phenotype of WT and mutant pollen suffers from a lack of explanations : the GC-MS profile of poaceatapel esters is based on an extracted ion chromatograph at m/z 189 as it is the case for poaceatapel from a saponified fraction (containing free triterpenes), whereas the mass spectra of the poaceatapel esters of C16 :0, C18 :0 and C18 :1 does not show such an ion (Supp. Figure 10).

Response:

Thanks for the comment. Sorry for missing label of m/z 189 on the mass spectra of the poaceatapel esters of C16 :0, C18 :0 and C18 :1. The m/z 189 is one of the characteristic fragment ions, and in the Supplemental Figure 11 of the revised manuscript was characteristic fragment ion m/z 189 was labelled.

Additional comments are listed below regarding supplementary figures and tables.

Typos (some).

Line 91, 112. fraction, not Fraction.

Line 104. polypoda-7,13,17,21-tetraene-3-beta-ol (not polypoda-7,13,17,21-tetraene).

Line 114. Affect (not effect).

Line 272. Triterpene esters (not ester).

Response:

Many thanks for the corrections. All above typos have been corrected.

Supplementary file.

Supplementary Figures and Tables

Line 290. Supplementary Figure 4. Are the seed set data shown in a panel (b) recorded at 30-60% RH ?

Response:

Yes, the seed set data shown in a panel (b) recorded are at 30-60% RH .The growth condition (30-60% RH) was added in legend of Supplementary Figure 4 in the revised version.

Line 302. Supplementary Figure 7. These pieces of data correspond to the chemical analysis of pollen coat described line 139-143 (extraction of 200g of pollen grains). I believe there is a concern with the GC diagrams (TIC or EIC) shown in panel b. In fact, the GC runs last 47

minutes as indicated on the horizontal axis of the chromatograph. The oven temperature program (lines 132-135) indicates that the peak framed as a triterpene ester at 39 min is eluted at 340°C (at the start of the last 8' plateau of 340°C). It would be interesting to see the scan m/z 50-800 of that peak by zooming in (whole mass spectrum of the peak), not only the EIC at m/z 189.

Response:

We re-analysed the gas chromatograph and mass spectrum of pollen coat extracts shown in Supplementary Figure 8a. We could not find any significant difference between whole mass spectrum of wild-type and mutant *E157*. So, the extract ion chromatogram at m/z 189 between 33-39 minutes was zoomed in the Supplementary Figure 8a.

In this experiment if the peak at 39 minutes is a triterpene ester it is not clear to me whether it could be a C16:0, or C18:0, or C18:1 ester of poaceatapol.

Response:

There are three peaks which differ between the chromatograph of mutant *E157* and WT. Their accurate mass spectra (Supplementary Figure 11) proposed that they are C16:0, C18:0, and C18:1 esters of poaceatapol.

In addition, what is the peak at 27 minutes present in WT and *E157* on the TICs at EICs m/z 189? Also, some peaks between 20-22 minutes in the mutant *E157* are increased (EIC m/z 189). What are their mass spectra?

Response:

The peak at 27 minutes (corresponding to the peak at 23 minutes in Supplementary Figure 8a of the revised manuscript) is the internal standard. The peaks between 20-22 minutes (corresponding to the peak at 16-18 minutes in Supplementary Figure 8a of the revised manuscript) are from several phytosterols and their mass spectra were given in Supplementary Figure 8d.

Line 326. Supplementary Figure 8. This shows the structural elucidation of the bicyclic triterpene of rice pollen coat as poaceatapol. Line 329 this compound should be written « polypoda-7,13,17,21-tetraene-3-beta-ol », not polypoda-7,13,17,21-tetraene. In (a), a compound is collected by semi-preparative LC for a further NMR analysis. Would it be possible to have the NMR spectrum as a supplementary file?

Response:

Yes, Line 104 in the revised manuscript and line 368 in Supplementary Information, the compound was written as “polypoda-7,13,17,21-tetraene-3-beta-ol”. The complete NMR spectra (¹H, ¹³C, and 2D) were added in Supplementary File 1.

In (b), the authors mentioned a reference standard of poaceatapol. What is the origin of the standard compound?

Response:

The NMR data of reference standard of poaceatapelol was from the literature (Ngyen and Harrison 1994 *Phytochemistry* 50: 471) and was not present in the revised manuscript.

In (c), the mass spectra data (scan m/z 50-700) should be presented, there is a peak on the TIC of SnOsOCS12 expressed in *Pichia*, it would have been easy to zoom in. In (d), the referee is confused with the whole data set. In fact, if the mass spectrum shown in (d) corresponds to the peak produced in *Pichia* as shown in (c) after extraction of the triterpene and GC-MS analysis of TMS-derivatives, how is the fragmentation explained with respect to the ion $m/z=189$? Please clarify the mass spectral data. The fragmentation of polypoda-7,13,17,21-tetraene-3-beta-ol as a TMS derivative should be drawn.

Response:

Yes, the mass spectra of TMS-derivative of OsOSC12 product in *Pichia* and poaceatapelol standard purified from rice panicles are given in Supplementary Figure 9d and 9e, respectively. The MS fragmentation of poaceatapelol is provided in Supplementary Figure 9f, indicating the m/z 189 is one of the characteristic fragment ions, and the extracted ion chromatograph of m/z 189 (EIC189) was shown.

Line 337. Supplementary Figure 9. Please specify the type of GC-MS analysis that is performed. It is important to figure out what is the m/z 189 ion to understand that poaceatapelol is present as a fatty acid ester in the pollen coat.

Response:

Yes, the type of GC-MS analysis is Agilent 7890/5977A GC/MS using electron ionization (line 130-133). The m/z 189 ion is a characteristic ion of poaceatapelol esters (see Supplementary Figure 11 in revised Supplementary Information).

Line 340. Supplementary Figure 10. The authors should indicate if this is a zoom of the TICs shown in Supplemental Figure 7. What is the ionization energy employed to generate ions shown here? What is the link with m/z 189?

Response:

Yes, Supplementary Figure 11 containing mass spectra of three poaceatapelol esters in revised Supplementary Information was from GC peaks in Supplementary Figure 8. It has been indicated in the figure legends. The applied ionization energy here is 70.0 eV. The m/z 189 ion was labelled in mass spectra of trietrapene poaceatapelol ester in revised supplementary Figures 11.

Methods

Line 3. Growth room conditions. It is not written whether RH is controlled or not in growth room. The authors propose that 30-60% RH is normal conditions but in my opinion 30% is a value that should be defined as dry. What is the device that enables RH measurement in growth rooms ?

Response:

The relative humidity in growth room was not controlled. This experiment was performed during the winter in Beijing, where the outside temperature is lower than 5°C and RH is lower than 30%. There are heating system in the growth room and the temperature is between 25-35°C, the RH was between 30-60% (Line 3 in Supplementary Information).

Line 7. Rice TILLING mutants. Selected mutants were backcrossed to WT cv Zhonghua 11. How many backcrosses ? Three BCs should clean 87.5% of the background, in principle.

Response:

Yes, all TILLING mutants were backcrossed to cv. Zhonghua 11 three times.

Line 22. qRT-PCR analysis of OsOSC12. The size of the amplicon obtained after qRT-PCR should be given (instead of « a portion »). Please indicate the OsGENE number for the Actin reference.

Response:

Yes, the size of the amplicon and gene number for Actin reference were added in the revised methods (Line 22).

Line 40. Anti-OSC12 antibody. What is the reason for choosing the 3-203 protein fragment ?

Response:

Based on the conformational disorder and epitope, Pfam domain and hydrophobic/hydrophilic of the protein, four recombination protein fragments (3-203, 217-460, 629-785, and 582-782) was chosen, and purified for production antibody. The antibody obtained from the mouse immured with the hydrophilic 3-203 protein fragment has the highest specificity to OsOSC12 protein in Western blot analysis.

Lines 100-103. Tilling experiment. The size of the mutagenesis experiment as number of M₁ and size of pooled M₂ should be given.

Response:

Yes, the information has been given in the revised method as the bellow: the 4950 fertile plants were generated from initial 6786 M₁ plants (line 102 in Supplementary Information). And DNA samples from 8 individual M₂ plants were pooled and submitted to TILLING analysis.

Line 130. Analysis of pollen coat. Is it a high temperature GC-MS protocol that is described here ?

Response:

Yes, the high temperature GC-MS protocol was used for analysis of triterpene esters (Line130-138, Supplementary Information).

Line 137. Fatty acid analysis of pollen coat extracts (PCE) obtained (line 128) by soaking pollen in organic solvents. Does it mean that pollen coat extracts described in lines 128-130 contain free fatty acids ?

Response:

Yes, the pollen coat extracts (PCE) by using this method contain mainly free fatty acids, tripterene esters, and other lipids (see Supplementary Figure 8).

Line 154. Although there is a reference 21 in this sentence I think it is important to recall to the reader the type of solvent that was employed to prepare an extract here.

Response:

Yes, a briefly extraction protocol has been added (line 155-157).

Line 156. The GC column has a size of 29,5 meters. Was it a 30-meter column shortened for normal maintenance of the device ? If this is the case I think it better to write 30-meter according to the manufacturer descriptions.

Response:

Yes, the GC column size is 30 meters and has been revised (line 132).

Lines 161-165. An analysis of triterpenes in the non-saponified and the saponified extracts suggest that triterpenes are present in the same biological fractions as 3-hydroxy- free compounds and as conjugates of fatty acids that are hydrolyzed in alkaline conditions. If triterpenes are analyzed as TMS-derivatives with MSTFA, why should the GC conditions be different ? This is misleading.

Response:

Yes, to erase this misleading, we re-analyzed triertpene poaceatapelol from all samples by use of the same GC-MS platform under the identical GC-MS protocol (DB-5HT column (30 m x 250 µm internal diameter, 0.1µm film, etc, see line 157-163 methods in Supplementary Information).

Lines 165-180. Large scale extraction and analysis.

The non saponifiable and fractionated hexane extract of S9-S11 panicles (6 kg) was purified on an Agilent 1200 LC semi-preparative instrument and fractions were monitored by GC-MS to look for a triterpene of interest. What is the fraction that gave a single peak in GC-MS ? There should be one since the authors state (line 172-173) that « the purified compound was directly introduced in the mass spectrometer for M^+ estimation, here the spectrometer being an Agilent Q-TOF 6540.

Response:

The fraction between 12-13 minutes gave the single peak in GC-MS analysis (see line 172-174 in Supplementary Information).

In addition, mass data collected from m/z 50-1000 could have shown the expected M^+ for the fatty esters of triterpene (eg M^+ 426 ($C_{30}H_{50}O$, triterpene), $^+ M^+$ 280 (linoleate) = 706). In this case I don't understand why the extract was saponified at the beginning of the whole analytical process, because the exact nature of triterpene esters is then lost.

Response:

Initially, we aimed to characterize the catalytic function of OsOSC12, which was predicted to produce triterpene aglycone. The saponified extract method was generally used for this type of analysis, and to identify its direct products.

Line 201. *Pichia pastoris* expression. What is the peculiarity of the strain X33 ? is it a wild-type strain ?

Response:

Yes, X33 is a wild-type *Pichia pastoris* strain, which is applied in Zeocin resistant expression vectors (line 228).

Line 221. Please write « ...were frozen in a high pressure freezing machine HPF Compact 01 (M. Wohlwend »

Response:

Yes, this has been corrected in the revised method (line 248).

Typos.

Line 167 : chromatography

Line 202 : derivatization

Line 203 : previously

Line 310, 311 : esters, not ester

Response:

Many thanks, these errors have been corrected in the revised method and figure legends.

Reviewer #3 (Remarks to the Author):

Line 82, At levels of 3.4 and 6.9 %, is the precision of the fold decrease values (4.93 and 4.51) statistically significant?

Response:

Thanks for the suggestions. These data were obtained from three biological replicates, and the statistical analysis was conducted. The *p*-value and s.d. were added in the revised manuscript (line 81-84).

Lines 98 Are the GC methods used in Supplement Figure 7 and Fig. 5 different? The WT EIC189 chromatograph in the former show baseline/noise between 23 and 26.5 min, whereas the peak of the target cpd and others falls into this retention time window. The peaks in the 38 to 42 minute window in Supplement Figure 7 seem to be congruent with those in Fig. 5 (c), but the baseline/noise level is different.

Response:

Sorry for this confusion. GC-MS analysis of pollen coat extracts were shown in the supplementary Figure7, and GC-MS analysis of anther extracts were shown in Figure 5.

We re-analyzed poaceatapelol esters from all samples by use of the same GC-MS platform under the identical GC-MS protocol (DB-5HT column (30 m x 250 μ m internal diameter, 0.1 μ m film, etc, see line130-138). The new chromatographs in Figure 5b show GC-MS analysis for anther extracts. The new chromatographs in Supplementary Figure 8a are GC-MS analysis for pollen coat extracts.

The isolation of 1 mg of polypoda-7, 13, 17, 21-tetraene from 6 kg of rice panicles is a major undertaking, but not described in any detail.

Response:

Yes, method of isolation of poaceatapelol was described in more detail in the Supplementary Information (line163-172) of the revised manuscript.

Figure 5. The structures of the poaceatapelols show a wedged bond for the methyl group attached to an olefinic carbon. This is incorrect as olefinic carbons are not chiral.

Response:

Thanks for this correction. The structure of poaceatapelol has been corrected (see Figure 5).

Supplement Figure 7. How was the quantification in (a) done? BY total GC-MS peak area? If so, (a) and (b) should be reversed. This also needs to be clarified.

Response:

Yes, the pollen coat extracts (PCE) have quantified by comparison of the total GC-MS peak area. And (a) and (b) were reversed (see Supplementary Figure 8 in the revised manuscript, which is corresponding to Supplementary Figure 7 of the previous version).

Which solvent was used to generate the extracts in (b)?

Response:

We had exchange the order of (a) and (b). The extract solvent used in (a) is ethyl ether.

No information is provided on the fractionation shown in (c); column material, conditions, yields, etc.

Response:

Yes, more information of column material, conditions and yields for separation of pollen coat materials was given in more detail in the legend (line354-361) of Supplementary Figure 8.

“Star” should be “asterisk”

Thanks for this correction and it has been replaced.

English language of the figure caption needs a major revision

Yes, figure caption were revised.

What does “Yellow square shows the difference” mean?

This sentence was removed.

Supplement Figure 8. The identification of the compound is solely based on ESI HR-MS and comparison of ¹³C NMR data reported in the literature. The typical 1D and 2D NMR data sets and NMR interpretations are not provided. Physicochemical characterization (UV, IR, alpha etc) are also missing.

Response:

Thanks for the comment. The complete NMR spectra (^1H , ^{13}C , and 2D) and physicochemical information (UV, IR, alpha) of the isolated compound (poaceatapel) were added in Supplementary File1 in the revised manuscript. Data interpretations were added in the revised Method Supplementary Information (lines 182-201). Based on this dataset, the structure of poaceatapel (polypoda-7,13,17,21-tetraene-3-beta-ol) was deduced.

Regarding the ^{13}C based identification of the compound: where was the reference standard obtained from? Or is this a case of confusion between a “reference standard” and a “literature reference”?

Response:

The NMR data of reference standard of poaceatapel was from the literature (Ngyen and Harrison 1994 *Phytochemistry* 50: 471) and was not present in the revised manuscript.

Assuming the authors intended to refer to a “literature reference”, the entire identification of the compound hinges on the quality of the assignments made in the 1998 Nguyen et al. publication. While it is likely that the compound isolated by both groups is the same (with the notable potential exception of the absolute configuration), due to the close congruence of the ^{13}C NMR data, the validity of the structural assignment made by Nguyen et al. should at least be reviewed critically. Preferably, and reflecting the importance of the isolated compound for the present study, the authors should deduce the structure ab initio, rather than assuming that the published structure is correct.

The data published by Nguyen et al. has largely incomplete ^1H NMR data and interpretations and lacks 2D NMR data completely. Moreover, Nguyen et al. provide no evidence about the absolute configuration and speculate that their compound “belongs to the normal series”. Using this as reference for establishing a structure is not acceptable. Importantly, for reproducibility, the authors should provide a complete set of spectroscopic and physicochemical data, as is standard in all chemistry and natural product journals.

Response:

Thanks for this suggestion. Yes, we deduced the structure of poaceatapel (polypoda-7,13,17,21-tetraene-3-beta-ol) based on a complete set of spectroscopic (^1H , ^{13}C , and 2D NMR data) and physicochemical data (UV, IR, $[\alpha]_D$) (see Supplementary File 1) that we obtained.

The comparison of the ^{13}C NMR data in (b) does not consider the different decimal precision (one or no decimal place vs. two decimal places). The Nguyen et al. reference lists all values with one decimal place.

Response:

Data from the Nguyen et al. reference was not present in the revised manuscript. The new ¹³C NMR data that we obtained in revised Supplementary Figure 9 has one decimal.

The intended word in the a/b/c footnote in (b) is probably “interchangeable”, and “assignment” should be pluralized.

Response:

Yes, these errors have been corrected.

The EI MS shown in (d) is missing interpretation. The description of the analyzed materials (“trimethylsilyl derivative from the yeast extraction expressing a 334 synthetic OsOSC12 with the codon optimization”) is unintelligible.

Response:

This figure legend was revised.

Linguistic Comments

While generally acceptable, English usage throughout the manuscript interferes unfavorably with scientific meaning. This happens in a relatively subtle way, but is important and requires careful revision. To give just one example, the second last sentence in the abstract is incomprehensible, mainly due to the word “sufficient” (necessary? essential?).

Response:

Yes, careful revision especially in supplementary information has been done. The statement in the abstract was revised.

Reviewers' comments:

Reviewer #1 (Remarks to the Author):

I appreciate the efforts from the authors in revising this manuscript to address my comments. In this revised version I still concerns:

- 1)The pollen coat data in Figure 4a and 4b and Supplementary Figure 6 and 7 appeared no obvious difference of the pollen coat between the WT and the mutants.
- 2)Figure 1 shows the expression of OsOSC12/PTS1 in tapetal cells, however, the authors did not describe clearly about the defect of tapetal cells in the mutants.

Reviewer #2 (Remarks to the Author):

The manuscript by Xue et al. « Deficiency of a triterpene pathway results in humidity-sensitive genic male sterility in rice » is a revision of a previous work submitted and reviewed earlier this year. After my inspection of the revised manuscript and supplemental files, and the detailed "response to reviewers" file, my opinion is that the authors have taken into account each point of criticism made by reviewers. Questions have been satisfactorily answered in the revised version of their work.

There are some typos.

Revised manuscript line 114 : microgram (not ug/mg)

Legend for Supplemental Figure 11 : « Samples » not « tSamples »

Supplemental method line 142 : delete « extract »

Supplemental method line 199 : cochinchinense should be in italics (not Vietnamese)

Reviewer #4 (Remarks to the Author):

This manuscript by Qi et al reports an interesting story about deficiency of a grass conserved triterpene synthase, OsOSC12/OsPTS1, results in humidity-sensitive genic male sterility in rice. The described indirect functions of triterpene poaceatapelol and its fatty acid esters are particularly interesting. The isolation and chemical characterization of poaceatapelol is one of major undertakings, as has been commented by other reviewers especially Reviewer #3.

The authors have addressed the major concerns of Reviewer #3 except for the absolute configuration issue. The planar structure of poaceatapelol appears to be correct from the Supplementary NMR spectra. I'm happy to see that the mistakenly transposed assignments of C-5 and C-9 in the literature (Nguyen and Harrison, *Phytochemistry*, 1998, 50: 471-476) have been corrected in Supplementary Figure 9(b), but cannot understand why there is still 4.8ppm difference for the chemical shift of C-8. In addition, the coupling constants in Supplementary Figure 9(b) need to be re-checked (eg. the coupling constant of H-3 is 8.8 Hz instead of 12.3Hz from the Supplementary ¹H NMR spectrum). It may not be easy to determine the absolute configuration of poaceatapelol since this compound is oily and only ca. 1 mg sample was obtained, but it would be better if the authors could measure a ROESY spectrum to confirm that 3-OH is indeed beta-oriented, and both C13(C14) and C17(C18) double bonds are in E form. Moreover, the optical rotation of poaceatapelol should be provided.

Supplementary Figure 8. What was the developing system for the TLC experiment in (c)? The ST fraction should have a relatively higher polarity than the TE fraction, but why the R_f value of ST fraction is larger than the TE fraction on silica gel TLC plate? In addition, the eluting system for the silica gel column chromatography is "hexane : ethyl ether = 20:1" in the figure legend, while in line

144 of Methods it is "hexane : dichloroethane = 10:1". Which one is correct? Moreover, I wonder if gradient eluents have been used for the column chromatography.

Methods. For the extraction, isolation and identification of poaceatapelol, the amounts of n-hexane extract and triterpene fraction should be provided (Lines 166-168). Semi-preparative HPLC is still ambiguous. It seems that a linear gradient mobile phase (from 90% to 100% methanol) was used in the first 20 min, but why another isocratic 100% methanol for as long as 45 min was used (Line 171)? The flow rate and the retention time of poaceatapelol should be provided as well. For the NMR measurement, ¹H was at 800 MHz while ¹³C was at 200 MHz (Lines 183-185). C₆D₆ was not just an internal standard, but more importantly an NMR solvent (Line 185), and "C₆H₆" in the same line should be revised to "C₆D₆". When expressing the chemical shift, subscript should be used for "deltaH" and "deltaC" (Lines 192-195). The NMR data of poaceatapelol are "similar to" those reported in the literature, rather than "same with" them (Line 198), especially when different NMR solvents were used.

Methods. Why the numbering of references starts from 26? And where is reference 21 cited in Line 230? Line 199, "cochinchinense" should be italic while "Vietnamese" should not.

Supplementary Figure 11. The molecular ion peaks of the three poaceatapelol fatty acid esters are very weak due to the limitation of electron ionization of mass spectrometry for long chain fatty acid analysis. If possible, an UPLC-ESI-MS/MS analysis for the TE fraction is suggested, so that their molecular ions could be better achieved, and the fragmentations could also be obtained to further support the existence of these fatty acid residues.

Main text. The three major fatty acids in the pollen coat extract were 16:0, 18:0 and 18:3 (Line 92), while the fatty acid residues of the major triterpene esters were 16:0, 18:0 and 18:1 (Line 112). How to explain why this should happen? Lines 115-118, how poaceatapelol and its derivatives indirectly affect pollen dehydration should be explained and discussed in more detail since these triterpenes are so critical in this work.

Reviewers' comments:

Reviewer #1 (Remarks to the Author):

I appreciate the efforts from the authors in revising this manuscript to address my comments. In this revised version I still concerns:

- 1) The pollen coat data in Figure 4a and 4b and Supplementary Figure 6 and 7 appeared no obvious difference of the pollen coat between the WT and the mutants.

Responds:

In Figure 4a and 4b, the pollen coat materials (the dark areas) in the bacula layer (between the tectum and the nexine) of WT pollen grains were indicated by **red arrows**, while it was not observed in the pollen of mutant *E157*. In Supplementary Figure 6 and 7, the pollen coat materials are indeed not obvious due to that area of the bacula layer were not zoomed in, but pictures are zoomed out to show whole pollen wall structures and to demonstrate the changes of lipid body during development of tapetum and pollen wall.

- 2) Figure 1 shows the expression of OsOSC12/PTS1 in tapetal cells, however, the authors did not describe clearly about the defect of tapetal cells in the mutants.

Responds:

We did not observe substantial deficiency of mutant tapetal cells other than the delayed degradation of lipid body. This change was described in the main text Line 121-123.

Reviewer #2 (Remarks to the Author):

The manuscript by Xue et al. « Deficiency of a triterpene pathway results in humidity-sensitive genic male sterility in rice » is a revision of a previous work submitted and reviewed earlier this year. After my inspection of the revised manuscript and supplemental files, and the detailed "response to reviewers" file, my opinion is that the authors have taken into account each point of criticism made by reviewers. Questions have been satisfactorily answered in the revised version of their work.

There are some typos.

Revised manuscript line 114 : microgram (not ug/mg)

Legend for Supplemental Figure 11 : « Samples » not « tSamples »

Supplemental method line 142 : delete « extract »

Supplemental method line 199 : cochinchinense should be in italics (not Vietnamese)

Responds:

Many thanks for the corrections. All above typos have been corrected.

Reviewer #4 (Remarks to the Author):

This manuscript by Qi et al reports an interesting story about deficiency of a grass conserved triterpene synthase, OsOSC12/OsPTS1, results in humidity-sensitive genic male sterility in rice. The described indirect functions of triterpene poaceatapelol and its fatty acid esters are particularly interesting. The isolation and chemical characterization of poaceatapelol is one of major undertakings, as has been commented by other reviewers especially Reviewer #3.

The authors have addressed the major concerns of Reviewer #3 except for the absolute configuration issue. The planar structure of poaceatapelol appears to be correct from the Supplementary NMR spectra. I'm happy to see that the mistakenly transposed assignments of C-5 and C-9 in the literature (Nguyen and Harrison, *Phytochemistry*, 1998, 50: 471-476) have been corrected in Supplementary Figure 9(b), but cannot understand why there is still 4.8ppm difference for the chemical shift of C-8.

Responds:

We have re-checked all NMR spectra data of this compound. We found that data assignment for C-8 (δ_C 135.2) in our manuscript is in consistent with that of *Org Biomol Chem* (2014) 12:3836 (δ_C 135.3). It is likely that assignment for C-8 with δ_C 140.0 in *Phytochemistry* (1998) 50:471 is incorrect. So this literature (*Phytochemistry* (1998) 50: 471) was not included in the revised manuscript.

In addition, the coupling constants in Supplementary Figure 9(b) need to be re-checked (eg. the coupling constant of H-3 is 8.8 Hz instead of 12.3Hz from the Supplementary ¹H NMR spectrum).

Responds:

The NMR data have been re-checked. For H-3, the signal should be a *dd* with *J* values at 11.2 Hz. and 2.7 Hz. This has been revised in the Supplementary Figure 9b.

It may not be easy to determine the absolute configuration of poaceatapelol since this compound is oily and only ca. 1 mg sample was obtained, but it would be better if the authors could measure a ROESY spectrum to confirm that 3-OH is indeed beta-oriented, and both C13 (C14) and C17 (C18) double bonds are in E-form.

Responds:

Many thanks for the suggestion. The stereochemistry of this compound was determined by the ROESY spectrum in combination with analysis of coupling

constant values of ^1H NMR and chemical shift values of ^{13}C NMR. It should be noted that H-3 resonated as a doublet of doublet with J values at 11.2 and 2.7 Hz in this compound. The large J value of 11.2 Hz indicated that H-3 was axially α -oriented, and accordingly the hydroxyl group at C-3 was β -oriented. The NOE correlations (Fig. 9b) of H-3/H₃-23 and H-3/H-5 also supported the above conclusion. The NOE correlations of H-13/H₂-15 and H-17/H₂-19 indicated that both of the Δ^{13} and Δ^{17} double bonds were E -configured. The relatively up-field shifted resonances for C-27 (δ_{C} 16.3) and C-28 (δ_{C} 16.1) in ^{13}C NMR also suggested that they were *syn* to C-12 and C-16, respectively, because of the γ -gauche effect, which also supported that both of the Δ^{13} and Δ^{17} double bonds were E -configured (Supplementary Information Line196-222).

Moreover, the optical rotation of poaceatapelol should be provided.

Responds:

The optical rotation of poaceatapelol ($[\alpha]_{\text{D}}^{25} = -8.2$ ($c = 0.05$, CH_3OH) was provided in the supplementary information (Line 228-230).

Supplementary Figure 8. What was the developing system for the TLC experiment in (c)? The ST fraction should have a relatively higher polarity than the TE fraction, but why the R_f value of ST fraction is larger than the TE fraction on silica gel TLC plate?

Responds:

Thanks for your comments. We used the solvent system hexane: dichloromethane = 2:1 in TLC (Supplementary Figure 8c). We re-check the GC-MS analysis of "ST fraction" and find that those fractions are the dehydrated sterols instead of sterols. So ST fractions were revised into DS (dehydrated sterols) fraction (this has been revised in the manuscript and the supplementary files).

In addition, the eluting system for the silica gel column chromatography is "hexane : ethyl ether = 20:1" in the figure legend, while in line 144 of Methods it is "hexane : dichloroethane = 10:1". Which one is correct? Moreover, I wonder if gradient eluents have been used for the column chromatography.

Responds:

Thanks for your comments. We indeed used a gradient eluents system including the dichloromethane from 0-10 % for 2 CV (column volume), 10-20 % for 10 CV and 20-100 % for 2 CV in hexane to elute the 20g silica gel column chromatography. Each fraction was collected in 20 ML test tube for TLC. These have been revised in the supplementary information (Line 142-146).

Methods. For the extraction, isolation and identification of poaceatapelol, the amounts of n-hexane extract and triterpene fraction should be provided (Lines 166-168). Semi-preparative HPLC is still ambiguous. It seems that a linear gradient mobile

phase (from 90% to 100% methanol) was used in the first 20 min, but why another isocratic 100% methanol for as long as 45 min was used (Line 171)? The flow rate and the retention time of poaceatapel should be provided as well.

Responds:

Yes, thanks for your suggestions. The amount of *n*-hexane extract is 52.23 g and the amount of triterpene fraction is 9.1 mg. The flow rate of semi-preparative HPLC for purification is 2.5 ml/min and the retention of time of poaceatapel is 28.48 min. These data have been provided in the revised supplementary information. After a linear gradient system, we used another isocratic 100% methanol for 45 min to wash the column.

For the NMR measurement, ¹H was at 800 MHz while ¹³C was at 200 MHz (Lines 183-185). C₆D₆ was not just an internal standard, but more importantly an NMR solvent (Line 185), and “C₆H₆” in the same line should be revised to “C₆D₆”. When expressing the chemical shift, subscript should be used for “deltaH” and “deltaC” (Lines 192-195).

Responds:

Yes, thanks for your correction. The sentence was changed as “¹H- and ¹³C-NMR (800 Hz) and 2D NMR (HMQC, HMBC, ROESY, and COSY) were measured, using C₆D₆ as an NMR solvent and internal standard.” (Line 186-188) and the “C₆H₆” was revised in to “C₆D₆”. The “deltaH” and “deltaC” was revised as “δ_H” and “δ_C” (Lines 196-222).

The NMR data of poaceatapel are “similar to” those reported in the literature, rather than “same with” them (Line 198), especially when different NMR solvents were used.

Responds:

This literature is not included in the manuscript, and this part was re-wrote (Line 196-222).

Methods. Why the numbering of references starts from 26? And where is reference 21 cited in Line 230? Line 199, “cochinchinense” should be italic while “Vietnamese” should not.

Responds:

The references in Supplementary Information were ordered to continue with that in main text, which is end in the 25th. The reference 21 was listed in the main text Line 187. This literature (*Phytochemistry* (1998) 50:471) was removed in revised manuscript.

Supplementary Figure 11. The molecular ion peaks of the three poaceatapelol fatty acid esters are very weak due to the limitation of electron ionization of mass spectrometry for long chain fatty acid analysis. If possible, an UPLC-ESI-MS/MS analysis for the TE fraction is suggested, so that their molecular ions could be better achieved, and the fragmentations could also be obtained to further support the existence of these fatty acid residues.

Responds:

Normally, triterpene and its ester are low polar compounds and not easy to be ionized in ESI model, that led to low sensitivities in ESI mass spectrometry. In addition, we found TE fraction has very low solubility in the solvent (e.g., methanol), which is normally as the solvent to dissolve samples in reverse phase HPLC system. The molecular ion peaks of the three poaceatapelol fatty acids in the enlarged figures are clear enough to elucidate their molecular weights.

Main text. The three major fatty acids in the pollen coat extract were 16:0, 18:0 and 18:3 (Line 92), while the fatty acid residues of the major triterpene esters were 16:0, 18:0 and 18:1 (Line 112). How to explain why this should happen?

Responds:

We found the amounts of triterpene and triterpene ester (0.31 µg/mg) are quite lower than that of free fatty acids (1-3 µg/mg) in pollen coat. We think that the three free fatty acids (16:0, 18:0 and 18:3) are not directly derived from the residues of triterpene esters. There is no consistent one-to-one match between the type of fatty acids in pollen coat and the fatty acid residues of triterpene esters.

Lines 115-118, how poaceatapelol and its derivatives indirectly affect pollen dehydration should be explained and discussed in more detail since these triterpenes are so critical in this work.

Responds:

The poaceatapelol and its derivatives affect pollen dehydration through impacting on transportation of fatty acids from lipid body to pollen wall. Poaceatapelol or its derivatives could interact with the membrane of lipid body, controlling the degradation process. Alternatively, they could act as signal molecules in the regulation of metabolites transportation. A brief discuss was added in the main text (Line 125-129). Its mechanism maybe complicated since we have identified more than 5 new HGMS mutants. The function of poaceatapelol and its derivatives in pollen coat formation could be elucidated through cloning these mutant genes.

Reviewers' comments:

Reviewer #4 (Remarks to the Author):

In the revised version of NCOMMS-16-30235A, the authors have made corresponding revisions and explanations to address my comments. Here are a few minor points that the authors should consider.

- 1) The provided optical rotation of poaceatapetol is in negative (-8.2), which is contrary to the literature value (+3.8) for (13E,17E)-polypoda-7,13,17,21-tetraen-3b-ol (Phytochemistry, 1998, 50:471). I wonder if poaceatapetol is an enantiomer of (13E,17E)-polypoda-7,13,17,21-tetraen-3b-ol? It would be perfect if the authors could further clarify this.
- 2) Reference 34 only contains the NMR data of poaceatapetol acetate, therefore I would suggest that the original reference (Phytochemistry, 1998, 50:471) should be included as well.
- 3) Methods. Line 144-146: Changed "a gradient eluents system including the dichloromethane from 0-10 % for 2 CV (column volume), 10-20 % for 10 CV and 20-100 % for 2 CV in hexane" to "a gradient eluents system including dichloromethane in hexane: 10% for 2 CV (column volume), 20 % for 10 CV and 100 % for 2 CV".
- 4) Legend of Supplementary Figure 9. Line 390: Change "NOE" to "NOESY".

Responds to Reviewer #4:

In the revised version of NCOMMS-16-30235A, the authors have made corresponding revisions and explanations to address my comments. Here are a few minor points that the authors should consider.

1) The provided optical rotation of poaceatapelol is in negative (-8.2), which is contrary to the literature value (+3.8) for (13E,17E)-polypoda-7,13,17,21-tetraen-3b-ol (Phytochemistry, 1998, 50:471). I wonder if poaceatapelol is an enantiomer of (13E,17E)-polypoda-7,13,17,21-tetraen-3b-ol? It would be perfect if the authors could further clarify this.

Responds: Thanks for raising this question. To address this question, we re-analyzed the optical rotation of poaceatapelol using the same solvent (chloroform) as the literature. The value is also positive (+7.25, $c = 0.035$, CHCl_3). The difference between two values (+7.25 vs +3.8) maybe due to that the different concentrations of samples were used for optical rotation assays. The value of optical rotation of poaceatapelol has been revised in the revised manuscript (Methods, Line 399-400).

2) Reference 34 only contains the NMR data of poaceatapelol acetate, therefore I would suggest that the original reference (Phytochemistry, 1998, 50:471) should be included as well.

Responds: Yes, the literature (Phytochemistry, 1998, 50:471) was included in the revised manuscript (Methods 389-393, Reference Line 530-531).

3) Methods. Line 144-146: Changed “a gradient eluents system including the dichloromethane from 0-10 % for 2 CV (column volume), 10-20 % for 10 CV and 20-100 % for 2 CV in hexane” to “a gradient eluents system including dichloromethane in hexane: 10% for 2 CV (column volume), 20 % for 10 CV and 100 % for 2 CV”.

Responds: Thanks for this suggestion and this sentence has been corrected (Methods Line 322-323).

4) Legend of Supplementary Figure 9. Line 390: Change “NOE” to “NOESY”.

Responds: Thanks for this correction. We have performed ROESY (supplementary Data 1) and “NOE” has been replaced by “ROESY” (Supplementary Figure 11 Line 129).

REVIEWERS' COMMENTS:

Reviewer #4 (Remarks to the Author):

The authors have made corresponding revisions to address all my questions, and I do not have any other additional comments.